# Pairwise Calibrated Rewards for Pluralistic Alignment

**Daniel Halpern**[1]    **Evi Micha**[2]    **Ariel D. Procaccia**[1]    **Itai Shapira**[1]
[1]Harvard University    [2]University of Southern California

## Abstract

Current alignment pipelines presume a single, universal notion of desirable behavior. However, human preferences often diverge across users, contexts, and cultures. As a result, disagreement collapses into the majority signal and minority perspectives are discounted. To address this, we propose reflecting diverse human preferences through a distribution over multiple reward functions, each inducing a distinct aligned policy. The distribution is learned directly from pairwise preference without annotator identifiers or predefined groups. Instead, annotator disagreements are treated as informative soft labels. Our central criterion is *pairwise calibration*: for every pair of candidate responses, the proportion of reward functions preferring one response matches the fraction of annotators with that preference. We prove that even a small outlier-free ensemble can accurately represent diverse preference distributions. Empirically, we introduce and validate a practical training heuristic to learn such ensembles, and demonstrate its effectiveness through improved calibration, implying a more faithful representation of pluralistic values.

## 1  Introduction

The alignment problem focuses on guiding AI systems to act in ways compatible with human values and intentions. At a high level, current methods steer models toward desired behaviors using curated sets of human preferences that capture behaviors that humans consider desirable or appropriate. The most successful approach is reinforcement learning from human feedback (RLHF) [1], which first trains a reward model—typically via the Bradley-Terry (BT) framework [2], on pairwise-preference data [3, 4]—and then fine-tunes a pre-trained model to align with the learned reward signal.

Implicit in current RLHF implementations is the assumption of a shared human intuition—a common ground among evaluators about what constitutes desirable behavior. While this assumption may hold for alignment objectives such as ensuring model safety, it generally does not apply to tasks where interpretations of "right" behavior inherently diverge across backgrounds, cultures, and beliefs [5–8]. BT-based reward models compress these diverse inputs into one scalar, blurring conflicting viewpoints into a one-size-fits-all model. This aggregation leads to a misspecified objective [9, 10] that struggles when faced with plural or contradictory feedback, marginalizing minority perspectives and failing to capture the full spectrum of human values [11–14]. Once such a reward is learned, algorithms like PPO [15] then push the policy to maximize this signal, triggering *preference collapse* [16], where majority views are further amplified, and response diversity is reduced [17–21].

To address these shortcomings, we replace the single-reward assumption with a *distribution* over reward functions; each independently assigns desirability scores to responses and they collectively span plausible interpretations of human judgment. By fine-tuning a distinct LLM for each reward in the support, the approach yields a distribution over policies. At inference time, these policies can be used in several ways [22, 23]: present a range of viewpoints or fold them into one inclusive answer, let users pick the policy that matches their taste, or sample responses directly from the distribution.

A straightforward approach to derive a distribution over reward functions is to partition annotators into predefined groups—such as demographic segments—or to cluster them by the similarity of

39th Conference on Neural Information Processing Systems (NeurIPS 2025).

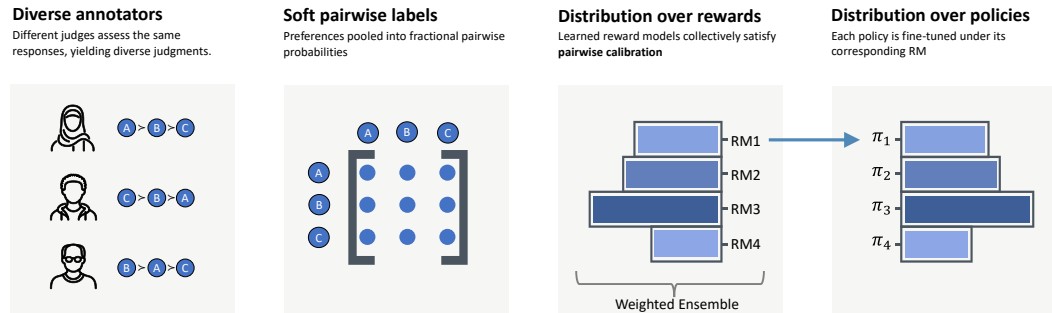

**Diverse annotators**
Different judges assess the same responses, yielding diverse judgments.

**Soft pairwise labels**
Preferences pooled into fractional pairwise probabilities

**Distribution over rewards**
Learned reward models collectively satisfy **pairwise calibration**

**Distribution over policies**
Each policy is fine-tuned under its corresponding RM

Figure 1: We explicitly model human preferences as a distribution over reward functions without relying on predefined annotator groups or identities. By enforcing pairwise calibration, this distribution faithfully captures the inherent diversity in aggregate human judgments. Each calibrated reward function then induces a distinct policy.

their recorded preferences [9, 24–26]. Such methods implicitly assume that each group holds a stable, coherent preference vector that carries over from one context to another. Human preferences, however, are fluid and highly context-dependent [27]; demographic or ideological attributes explain only part of the observed variation, and annotators who match on all recorded traits can still disagree sharply [28]. Moreover, overly fine-grained groups may lack enough data for reliable training.

Without explicit annotator identification or predefined groups, learning a distribution over rewards requires disentangling the hidden preference dimensions inside aggregated feedback. Concretely, rather than associating preferences with stable annotator identities, we look for a *small* set of internally consistent reward functions—an *ensemble*—that together explain conflicting human judgments across contexts. Our aim is to achieve a novel property we call *pairwise calibration*: for every pair of candidate responses, the share of reward functions in the support of the ensemble favoring one response matches the fraction of annotators who express that preference. A pairwise-calibrated ensemble gives each reward function a coherent viewpoint and its own policy. Taken together, these policies mirror a broader range of human preferences and avoid preference collapse (see Figure 1). Thus, pairwise calibration provides a principled mechanism to preserve pluralism, without imposing rigid clusters or relying on annotator tags.

**Our results.** In Section 2 we present our model and formally define *pairwise calibration*. Section 3 shows that while ensembles can trivially satisfy pairwise calibration when they are sufficiently large (e.g., at the scale of the number of annotators), constructing such ensembles is NP-hard (Theorem 1). More importantly, our main goal is to find practical ensembles with small support. In light of these observations, we show in Theorem 2 that an ensemble of size $O(1/\varepsilon)$ is sufficient to achieve an average calibration error of $\varepsilon$, and Theorem 3 further shows that including extreme outlier preferences is unnecessary, as an outlier-free ensemble remains nearly pairwise-calibrated. Moreover, Theorem 4 shows that achieving pairwise calibration can be learned with a limited number of pairwise comparisons, providing a generalization guarantee. Section 4 asks whether such ensembles can be efficiently constructed in practice and answers with a forward stagewise additive modeling (FSAM) procedure: each iteration trains a reward model on the current residual calibration error and re-weights it into the mixture, giving a tractable method for constructing a compact ensemble. Finally, Section 5 demonstrates that these ensembles attain lower calibration mean squared error (MSE) than the theoretically optimal single deterministic reward and that the reward models within each ensemble are diverse.

**Related work.** Our work contributes to the growing field of *pluralistic alignment*, training AI systems to accommodate the natural diversity of human values and perspectives [23, 29–32]. We briefly cover the broad approaches taken in this emerging area. A more comprehensive overview can be found in Appendix B.

Current approaches to capture preference diversity include:

*Modifying single scalar reward models*, such as adding regularizers or other algorithmic techniques, to ensure that the reward is not overly optimized to the majority [16, 33–35]. Nonetheless, methods reliant on a single scalar reward still risk oversimplifying genuine diversity in user opinions.

*Training multiple reward models for distinct population clusters or groups* [9, 24–26]. While more direct, this approach does depend on defining fixed groups and can struggle with data sparsity.

*Personalization, attempting to create a model optimized for each individual user* [36, 13]. To remain feasible, these methods must sometimes make strong assumptions on latent embeddings, or make use of partial fine-tuning. Furthermore, this approach still risks creating "echo chambers."

## 2 Our Approach

**Notation.** Let $\mathcal{X}$ be the context space and let $\mathcal{Y}$ be the response space. A reward function $r_\theta : \mathcal{X} \times \mathcal{Y} \to \mathbb{R}$ assigns a scalar value indicating the desirability of a response given a context, where $\theta \in \Theta$ is a set of parameters. A *k-ensemble* (or simply an ensemble) is a tuple $(r_{\theta_1}, \ldots, r_{\theta_k}; \alpha_1, \ldots, \alpha_k)$, where $\alpha_1, \ldots, \alpha_k$ are mixture weights forming a probability distribution over the reward functions (each $\alpha_j \geq 0$ and $\sum_{j=1}^{k} \alpha_j = 1$).

We denote by $\mathcal{N}$ the population of annotators, and we write $i \sim \mathcal{N}$ to mean drawing an annotator uniformly from the population. We assume each annotator $i \in \mathcal{N}$ has a preference relation $\succ_i$ such that for any context $x \in \mathcal{X}$ and distinct responses $y_1, y_2 \in \mathcal{Y}$, $y_1 \succ_i y_2 \mid x$ indicates that annotator $i$ prefers $y_1$ to $y_2$ given context $x$. We call a tuple $(x, y_1, y_2)$ a *response triple* or simply a *triple*. We assume the reward space is rich enough such that the annotators are *reward-inducible*: for each $i \in \mathcal{N}$, there is a reward $r_\theta$ such that $y_1 \succ_i y_2 \mid x$ exactly when $r_\theta(x, y_1) > r_\theta(x, y_2)$.

A policy $\pi(y \mid x)$ is a probability distribution over outputs $y$ conditioned on a context $x$. We assume there is an underlying distribution over contexts $D_\mathcal{X}$ and base policy $\pi_0$. Fine-tuned policies are typically learned by optimizing responses to maximize expected rewards according to the corresponding reward function $r_\theta$.

**Pairwise calibration.** Our key conceptual innovation is the requirement that the ensemble faithfully reflects human preference frequencies. Intuitively, an ensemble is *pairwise calibrated* if the pairwise comparisons it induces align with the proportion of human annotators who prefer one response over another. Formally, consider a context $x \in \mathcal{X}$ and two distinct candidate responses $y_1, y_2 \in \mathcal{Y}$. We define $p^\star(x, y_1, y_2)$ to be the true fraction of annotators that prefer $y_1$ to $y_2$ given context $x$:

$$p^\star(x, y_1, y_2) := \Pr_{i \sim \mathcal{N}} [y_1 \succ_i y_2 \mid x] \in [0, 1]. \tag{1}$$

Given an ensemble $\mathbf{r} = (r_{\theta_j}; \alpha_j)_{j \in [k]}$, denote the induced preference probability by

$$\hat{p}^{\mathbf{r}}(x, y_1, y_2) := \sum_{j=1}^{k} \alpha_j \cdot \mathbf{1}[r_{\theta_j}(x, y_1) > r_{\theta_j}(x, y_2)].^{[1]}$$

If $\mathbf{r}$ is clear from context, we may drop it from the $\hat{p}$ notation.

**Definition 1** (Pairwise calibration). *We say an ensemble $\mathbf{r}$ is $\varepsilon$-pairwise calibrated if:*

$$\mathbb{E}_{x \sim D_\mathcal{X}, y_1, y_2 \sim \pi_0(\cdot \mid x)} \left[ (\hat{p}^{\mathbf{r}}(x, y_1, y_2) - p^\star(x, y_1, y_2))^2 \right] \leq \varepsilon.$$

If an ensemble is 0-pairwise calibrated, we call it *perfectly* pairwise calibrated. We use squared error for analytical convenience, though other divergences that penalize discrepancies between $\hat{p}$ and $p^\star$ (e.g., absolute error or cross-entropy) could also serve this purpose. Calibration is defined here at the reward level based only on how they order the responses. This is not the only natural definition: pairwise calibration could instead depend directly on the probabilities induced by the learned policies. Our choice is primarily motivated by tractability and the observation that downstream policies are largely influenced by the ordering of rewards. See Appendix A for further discussion and justification of this design choice.

---

[1] We assume that chosen reward functions never produce ties. This is a minor assumption, as in continuous parameter spaces, the parameters that induce ties $r_\theta(x, y_1) = r_\theta(x, y_2)$ with $y_1 \neq y_2$ typically form a measure-zero set, so any perturbation almost surely removes them.

**Non-identifiability.** Once annotator identities are dropped, the only observable data are the pairwise preference probabilities $p^\star$ in Equation 1. Prior work [37–39] shows that the same $p^\star$ can be generated by infinitely many distinct mixtures of annotator-specific preferences. Because of this, recovering the "true" annotator preference distribution is information-theoretically impossible; our goal is therefore to learn *any* ensemble that is pairwise calibrated to $p^\star$, rather than to reconstruct hidden annotator groups.

**Policy ensemble inference.** A pairwise-calibrated ensemble induces a mixture of policies, $(\pi_1, \ldots, \pi_k)$, where $\pi_j$ is aligned to $r_{\theta_j}$; yet this alone does not tell us *how* to deploy that distribution at inference time. Below we outline three complementary approaches—inspired by Sorensen et al. [23]—that can be selectively employed to serve different pluralism purposes depending on the context. Together they offer a practical toolkit for pluralistic alignment.

In *balanced pluralism* mode, the system queries every LLM in the support. After sampling the resulting set of candidate outputs, it can either present them in an Overton-style slate, useful for open-ended advice, policy deliberation, and brainstorming; or distill them into a single consensus statement, aligning with prior work on consensus building [40, 41].

In *steerable pluralism* mode, the system, on a per-prompt basis, selects one policy $\pi_j$ whose persona or metadata best matches a declared user preference and produces that policy's output. The result is a single voice that fits personal assistants, brand-specific copy, or group-specific safety policies, without the overhead of user-specific fine-tuning or elaborate prompt engineering.

In *distributional pluralism* mode, the system samples a policy from the mixture in proportion to its weight and returns that policy's output. Applied over many requests, this procedure maintains population-level diversity and counteracts the tendency of aligned models to exhibit low output diversity by recycling a small set of high-reward responses [19–21]. This mode is especially valuable in creative workflows—text-to-image generation, story creation, and recommendation engines—where safety constraints must coexist with a rich variety of outputs.

**Small support.** We deliberately restrict the ensemble to a small support. A compact ensemble is easier to train, tune, and deploy, and it generalizes better (see Theorem 4). Furthermore, the inference methods discussed require storing and efficiently querying the full ensemble, constraining $k$ to stay relatively small. Finally, Section 3.1 shows that only $O(1/\varepsilon)$ reward functions suffice for $\varepsilon$-pairwise calibration, so increasing $k$ yields diminishing gains.

Beyond these practical reasons, we intentionally target a small support for normative considerations. Limiting the ensemble size mitigates societal risks inherent in extreme personalization—such as reinforcing existing biases, promoting echo chambers, and exacerbating polarization (see *personalization within bounds* [13]). Taken together, Theorems 2 and 3 show that a limited-support distribution over rewards preserves meaningful diversity without over-tailored personalization.

**Outlier reward functions.** While pairwise calibration ensures the ensemble *as a whole* faithfully reflects aggregate human preference frequencies, we are also interested in the behavior of the individual reward functions $r_{\theta_j}$ that constitute the ensemble. A well-calibrated ensemble could, in principle, still contain individual reward functions that represent extreme or undesirable viewpoints.

To quantify how much a single reward function $r_\theta$ deviates from the overall population preferences, we define its *disagreement score* as:

$$\Phi(\theta) = \Pr_{\substack{x \sim D_{\mathcal{X}}, y_1, y_2 \sim \pi_0(\cdot | x), \\ i \sim \mathcal{N}}} \left[ \mathbf{1}[r_\theta(x, y_1) > r_\theta(x, y_2)] \neq \mathbf{1}[y_1 \succ_i y_2 \mid x] \right].$$

The disagreement score measures how frequently the reward function disagrees with the population. More specifically, it is the probability that for a randomly selected annotator $i \sim \mathcal{N}$ and context response triple $(x, y_1, y_2)$, $r_\theta$ prefers one response while $i$ the other.

Of course, on populations with high amounts of diversity, it may be impossible for any reward to have particularly low disagreement score. Given this, we say a reward function $r_\theta$ is a $(\beta, \gamma)$-*outlier* if $\Phi(\theta) > \beta \cdot \inf_{\theta' \in \Theta} \Phi(\theta') + \gamma$, which normalizes disagreement relative to the best-achievable score.

## 3 Theoretical Guarantees

Having defined pairwise calibration, we now address three fundamental questions: (i) *Can approximately pairwise-calibrated ensembles be supported on a relatively small set of reward functions?*

(ii) *Can such calibration be achieved without using extreme outlier rewards?* (iii) *Do ensembles calibrated on a finite dataset generalize to unseen populations?* We answer all three in the affirmative.

## 3.1 Pairwise-Calibrated Ensembles with Small Support

A uniform mixture over the annotators' true reward functions would, in principle, achieve perfect pairwise calibration. However, this approach is impractical for two reasons. First, the true reward functions are never observed—only partial pairwise information provided. Second, this ensemble's support size would need to scale with the number of annotators, making the approach computationally infeasible.

For now, we set aside the second issue and focus on the first. We show that *finding* a perfectly calibrated ensemble is computationally hard even in an extremely simplified setting: there is only a single context, (i.e., $|\mathcal{X}| = 1$), $m$ possible responses (i.e., $|\mathcal{Y}| = m$), and we have access to the true fractions of annotators who prefer $y_1$ to $y_2$ for all pairs, (i.e., all of the values $p^\star(x, y_1, y_2)$).

Under these conditions, the set of pairwise comparisons we wish to calibrate to is finite. A straightforward application of Carathéodory's theorem guarantees that there exists a perfectly calibrated ensemble of size only $\Theta(m^2)$ (see Appendix C.1). Yet, even under these strong simplifying assumptions—and a guarantee that only a relatively small ensemble is required—it is computationally hard to find one.

**Theorem 1.** *If $P \neq NP$, then there does not exist a polynomial-time algorithm for finding a perfectly pairwise calibrated ensemble when $|\mathcal{X}| = 1$, $|\mathcal{Y}| = m$ for some finite $m$ and given access to $p^\star(x, y_1, y_2)$ for all $(x, y_1, y_2)$.*[2]

This setup is analogous to one in *social choice theory* [42]; see Appendix C.2 for details. At a high level, the proof (Appendix D.1) shows that if we could efficiently find perfectly calibrated ensembles, we would be able to determine whether certain values of $p^\star$ are realizable by *any* underlying distribution over reward functions. This provides a *membership oracle* to the set of realizable $p^\star$, which turn out to form a useful convex set known as the *linear ordering polytope*. Membership oracles allow us to optimize linear functions over the polytope, thereby enabling us, if we are careful with various approximations, to solve classic NP-hard problems such as *Minimum Feedback Arc Set*.

Next, we address the second consideration. We show that small-support ensembles can indeed achieve (approximate) pairwise calibration.

**Theorem 2.** *For any $\varepsilon > 0$, there exists a $O(\varepsilon^{-1})$-ensemble that is $\varepsilon$-pairwise-calibrated.*

This is shown (in Appendix D.2) using the probabilistic method: by sampling $O(1/\varepsilon)$ reward functions according to a particular strategy, the expected pairwise calibration error is at most $\varepsilon$, guaranteeing the existence of at least one ensemble meeting the criterion.

## 3.2 Pruning and Outlier Control

Matching aggregate preferences is not enough if some reward functions deviate so sharply from the population that they could endorse behaviors most annotators would strongly reject. The next question is whether achieving calibration *forces* us to include such extreme outliers. Using the previously defined *disagreement score* $\Phi(\theta)$ as our proxy for how far a reward strays from mainstream judgments, our next result demonstrates it is possible to remove these extreme outliers without sacrificing too much pairwise calibration.

**Theorem 3.** *Suppose there exists a $k$-ensemble $\mathbf{r}$ that is $\varepsilon$-pairwise calibrated. Then, for all $\beta \geq 2$, the total weight of reward functions in $\mathbf{r}$ that are $(\beta, (\beta+1) \cdot \sqrt{\varepsilon})$-outliers is at most $\frac{1}{\beta-1}$. Furthermore, there exists another $\ell$-ensemble (for $\ell \leq k$) $\mathbf{r}'$ that is $(\sqrt{\varepsilon} + \frac{1}{\beta-1})^2$-pairwise calibrated and does not contain any $(\beta, (\beta+1) \cdot \sqrt{\varepsilon})$-outliers, and this $\mathbf{r}'$ can be computed with access only to $\mathbf{r}$.*

This proof (Appendix D.3) begins with a Markov-style argument: most reward functions in $\mathbf{r}$ cannot deviate substantially from the ensembles aggregate predictions ($\hat{p}^{\mathbf{r}}$). Furthermore, we can bound the

---

[2]Formally, the algorithm takes as input the rational values $p^\star(x, y_1, y_2)$, encoded as pairs of binary integers, and must run in time polynomial in the size of this instance.

disagreement score of individual reward functions based on how much they deviate from $\mathbf{r}$ and the initial pairwise calibration of $\mathbf{r}$. Functions exceeding this bound are identified as outliers. We then demonstrate that removing these outliers—a process requiring comparison only against $\hat{p}^{\mathbf{r}}$ (not the true $p^{\star}$ values)—has minimal impact on the overall pairwise calibration.

## 3.3 Generalization

In practice, we do not have direct access to the true preference proportions $p^{\star}$. Rather, we typically work with a finite dataset $\mathcal{D}$ of pairwise comparisons. We assume the dataset has the following form: $\mathcal{D} = \{(x^{(i)}, y_1^{(i)}, y_2^{(i)}, p^{(i)})\}_{i=1}^{N}$, where each entry is generated by sampling a context $x \sim D_{\mathcal{X}}$, two candidate responses $y_1, y_2 \sim \pi_0(\cdot \mid x)$, and $n$ annotators $i_1, \ldots, i_n \sim \mathcal{N}$. The empirical preference is then set to $p = \frac{1}{n} \sum_{j=1}^{n} \mathbf{1}[y_1 \succ_{i_j} y_2 \mid x]$, the fraction of annotators who prefer $y_1$ over $y_2$.

Let $\mathcal{R}_k$ be the set of all $k$-ensembles. Our goal is to, using the dataset, find an ensemble $\mathbf{r} \in \mathcal{R}_k$ that achieves small pairwise calibration error on the true population:

$$\mathcal{L}(\mathbf{r}) = \mathbb{E}_{x \sim D_{\mathcal{X}}, y_1, y_2 \sim \pi_0(\cdot|x)} \left[ (\hat{p}^{\mathbf{r}}(x, y_1, y_2) - p^{\star}(x, y_1, y_2))^2 \right].$$

However, we only have access to the dataset, and the corresponding empirical loss:

$$\widehat{\mathcal{L}}(\mathbf{r}) = \mathbb{E}_{(x, y_1, y_2, p) \sim \mathcal{D}} \left[ (\hat{p}^{\mathbf{r}}(x, y_1, y_2) - p)^2 \right].$$

Using $\widehat{\mathcal{L}}$ to estimate $\mathcal{L}$ presents an inherent challenge: $\widehat{\mathcal{L}}$ is a biased estimator of $\mathcal{L}$.

**Lemma 1.** *For any ensemble $\mathbf{r} \in \mathcal{R}_k$, $\mathbb{E}_{\mathcal{D}}[\widehat{\mathcal{L}}(\mathbf{r})] = \mathcal{L}(\mathbf{r}) + C$, where*

$$C := \mathbb{E}_{x \sim D_{\mathcal{X}}, y_1, y_2 \sim \pi_0(\cdot|x), p \sim Bin(n, p^{\star}(x, y_1, y_2))/n} \left[ (p - p^{\star}(x, y_1, y_2))^2 \right].$$

This is proved in Appendix D.4. The term $C$ represents an irreducible error introduced due to only $n$ annotators responding per data point. Even a perfectly pairwise calibrated ensemble $\mathbf{r}$ will incur error $C$ in expectation over $\mathcal{D}$.

Despite this limitation, learning-theoretic tools allow us to provably estimate $\mathcal{L}$ up to the constant shift $C$. This implies that minimizing the empirical loss $\widehat{\mathcal{L}}$ remains a sound strategy for obtaining a model with optimal pairwise calibration.

To formalize this, let $\mathcal{F} = \{(x, y_1, y_2) \mapsto \mathbf{1}[r_\theta(x, y_1) \geq r_\theta(x, y_2)] \mid \theta \in \Theta\}$ be the class of binary comparison functions induced by the reward models, i.e., these are functions parameterized by $\Theta$, mapping triples $(x, y_1, y_2)$ to $\{0, 1\}$, outputting 1 exactly when $r_\theta(x, y_1) \geq r_\theta(x, y_2)$. Defining $\mathcal{L}'(\mathbf{r}) = \mathcal{L}(\mathbf{r}) + C$, we have:

**Theorem 4.** *Suppose $\mathcal{F}$ has finite VC-dimension $d$. Then, for any $\delta > 0$, with probability at least $1 - \delta$ over a dataset $\mathcal{D}$ of $N$ i.i.d. samples,*

$$\sup_{\mathbf{r} \in \mathcal{R}_k} \left| \mathcal{L}'(\mathbf{r}) - \widehat{\mathcal{L}}(\mathbf{r}) \right| \leq \sqrt{\frac{2d' \log(\frac{eN}{d'})}{N}} + \sqrt{\frac{\log \frac{1}{\delta}}{2N}},$$

*where $d' = 20(d+1)k \cdot \log(2(d+1)k) \in O(kd \log(kd))$.*

The proof (Appendix D.5) applies uniform convergence bounds for agnostic PAC learning. However, to do so requires bounding the pseudo dimension of the loss class $\{((x, y_1, y_2), p) \mapsto (\hat{p}^{\mathbf{r}}(x, y_1, y_2) - p)^2 \mid \mathbf{r} \in \mathcal{R}_k\}$. To bound this, we apply a sequence of transformations to $\mathcal{F}$, bringing it closer to the loss class, and show each step individually does not substantially increase the pseudo-dimension.

As a concrete example, suppose we have an embedding function $\varphi : \mathcal{X} \times \mathcal{Y} \to \mathbb{R}^\ell$ which maps context-response pairs into $\mathbb{R}^\ell$, and that the reward functions $\{r_\theta \mid \theta \in \Theta\}$ are linear functions over these embeddings: $r_\theta(x, y) = \theta \cdot \varphi(x, y)$ where $\Theta = \mathbb{R}^\ell$. This setup corresponds to the common practice of learning a reward model by removing the final layer of a pretrained language model (yielding an embedding function), attaching a new linear head that maps the embedding to a scalar reward, and training only this final layer on preference data. Here, the comparison functions become $\mathbf{1}[\theta \cdot \varphi(x) \geq \theta \cdot \varphi(y)] = \mathbf{1}[\theta \cdot (\varphi(x) - \varphi(y)) \geq 0]$ which correspond to linear classifiers over the embedding space. This is known to have VC-dimension at most $\ell$ [43].

# 4 Implementation via Residual Reward Calibration

Given that only a small mixture is theoretically sufficient, the practical question is: *can we construct it in practice?* In principle, we could learn an ensemble by solving the full optimization problem:

$$\min_{\substack{\alpha_1,\ldots,\alpha_k \\ \theta_1,\ldots,\theta_k}} \mathbb{E}_{(x,y_1,y_2)\sim\mathcal{D}}\left[\left(p(x,y_1,y_2) - \hat{p}(x,y_1,y_2)\right)^2\right].^3 \tag{2}$$

However, directly minimizing this objective requires a computationally intensive search over both the mixture weights and the reward model parameters, rendering it impractical.

Instead, we propose a heuristic approach based on *forward stagewise additive modeling (FSAM)* (see Hastie et al. [44] for an overview) that decomposes the problem into a sequence of more tractable subproblems. At each step, we fit a new reward model to the current residual error of the ensemble, keeping previously learned models fixed; pick a mixing weight that best reduces that error, and append the new model to the mixture. We observe that, despite its heuristic nature, this approach tractably recovers ensembles that are approximately pairwise-calibrated on real-world datasets.

We start by detailing how we train the first reward model (i.e., $k = 1$). Even in this case, Equation 2 already departs from vanilla RLHF in two ways. First, the target is the observed preference fractions, which we refer to as the *soft labels*. Even in cases where multiple annotators disagreed on which response is better, typical alignment datasets flatten these to binary labels $p(x,y_1,y_2) \in \{0,1\}$ by collapsing multiple annotators' votes into a single "majority-decision" label [45–51]. In fact, several high-profile alignment deployments train exclusively on these binary signals and treat them as a gold standard, discarding the underlying soft-label information as noise [52–54].[4] By contrast, we *need* access to this soft label data, as we aim to match the observed preference fractions. This training regime is conceptually distinct from methods that merely smooth hard labels—such as label smoothing [55, 56]—or from approaches that try to infer a latent consensus from soft labels [57, 58]. Second, we present our method in terms of the MSE loss, while typical RLHF uses cross-entropy. MSE aligns cleanly with later ensemble iterations, though the same construction could be carried out with a cross-entropy loss as well.

Having defined the $k = 1$ special case, we now turn to the rest of the ensemble. At iteration $j > 1$ we expand the current ensemble $\mathbf{r}_{j-1} = (r_{\theta_1},\ldots,r_{\theta_{j-1}};\alpha_1,\ldots,\alpha_{j-1})$ by adding a new reward model $r_{\theta_j}$. Let

$$\varepsilon_j(x,y_1,y_2) = p(x,y_1,y_2) - \hat{p}^{\mathbf{r}_{j-1}}(x,y_1,y_2)$$

be the residual calibration error. We fit $r_{\theta_j}$ by minimizing

$$\mathbb{E}_{(x,y_1,y_2)\sim\mathcal{D}}\left[\left(\varepsilon_j(x,y_1,y_2) - \sigma\left(r_{\theta_j}(x,y_1) - r_{\theta_j}(x,y_2)\right)\right)^2\right],$$

using the sigmoid $\sigma(z) = 1/(1 + \exp(-z))$ as a smooth proxy for the binary vote indicator.

Each reward model starts from the same supervised fine-tuned (SFT) checkpoint: we drop the token-prediction head, attach a freshly initialized single-node reward head, keep all other parameters frozen from SFT, and then fine-tune on preference pairs.[5] Once $r_{\theta_j}$ is trained, we add it to the ensemble, and reoptimize coefficients $(\alpha_1,\ldots,\alpha_j)$ to minimize the training MSE, given $r_{\theta_1},\ldots,r_{\theta_j}$.

The method incrementally concentrates capacity on the "difficult" comparisons—those with the largest residual magnitude. Residuals may be negative or exceed 1, applying even more pressure to the subsequent reward model to converge toward the soft labels.

The procedure can terminate after a fixed $k$ iterations or, alternatively, via early stopping when the validation loss stops improving. Each additional iteration adds one fine-tuning pass, so runtime grows linearly with the number of reward models.

# 5   Empirical Results

We now evaluate the FSAM procedure to test whether this heuristic can learn pairwise-calibrated ensembles on real alignment datasets. We focus on two questions: (i) *Can a small ensemble of weak reward models match the observed vote fractions more accurately than any single reward model?* (ii) *Do the individual reward models in the ensemble capture distinct preference patterns rather than duplicating one another?* For both, we find positive results.

**Datasets.**  To be suitable for our experiments, datasets must satisfy two conditions: (i) they must include—or allow us to reconstruct—pairwise comparisons between distinct LLM outputs; and (ii) every comparison must be rated by more than one annotator. We use four public datasets that satisfy these requirements and exhibit annotator disagreement: `MultiPref` [59], `PersonalLLM` [60], `HelpSteer2` [61], and `Reddit TL;DR` [62]. Summary statistics appear in Table 1; further details about the datasets, including pre-processing, can be found in Appendix F. These datasets vary substantially in both domain and size.

| Name | Pref. Pairs | Unique Prompts | Annotation | Avg # Annots. | Avg $p$ | |
|---|---|---|---|---|---|---|
| `MultiPref` [59] | 9,413 | 4,791 / 532 | Human annotators | 4.0 | | 63% |
| `PersonalLLM` [60] | 263,256 | 9,402 / 1,000 | Model-based scores | 10 | | 76% |
| `HelpSteer2` [61] | 21,000 | 10,000 / 1,000 | Human annotators | 3.5 | | 74% |
| `Reddit TL;DR` [62] | 3,217 | 729 / 845 | Human annotators | 7.56 | | 84% |

Table 1: Summary statistics for the four preference datasets (see Appendix F.2 for details). $p$ is defined here as the mean majority-agreement score, i.e., the average share of annotators who chose the majority option; by definition this share is at least $0.5$.

**Ensemble of weak reward models vs. an optimal single reward model.**  We fit an ensemble of k=8 reward models on each dataset, starting from supervised fine-tuned checkpoints of Meta-Llama-3-8B [63] (a juggernaut in the small-model bracket). We compare the ensemble to a *theoretically optimal single deterministic reward model* that always selects the majority-preferred answer for every comparison. Such a model minimizes binary error and its mean-squared calibration loss cannot fall below $\sum_i \min\{p_i^2, (1-p_i)^2\}$. A mixture of weak deterministic reward models offers a finer-grained set of outputs but, a priori, need not perform better.

Our results (Figure 2) show that, in many cases, an ensemble of only 2-4 such rewards already achieves noticeably better calibration on held-out prompts. Gains tend to taper off around six models.

---

[3]Throughout this section, we use $p(x, y_1, y_2)$ to denote the observed fraction of annotators that responded $y_1 \succ y_2 \mid x$.

[4]This choice is surprising because in the BT model, soft-label BCE matches the true log-likelihood, and the vote fraction reveals how small the underlying score gap is between the two responses (see Appendix E).

[5]Training details appear in Appendix F.1.

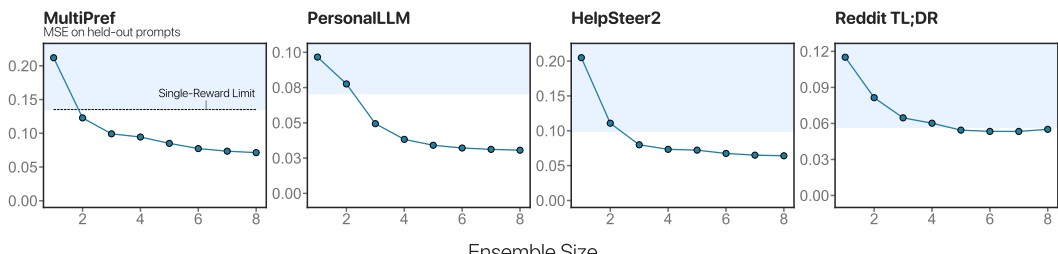

Figure 2: MSE calibration error on held-out prompts as a function of ensemble size $k$. Each curve is obtained by training a mixture of eight weak reward models with our FSAM procedure, which greedily adds one base model at a time to minimize the residual calibration error. The shaded band marks the floor for any single deterministic reward model, $\sum_i \min\{p_i^2, (1-p_i)^2\}$. Across all datasets, ensembles of only two to four rewards already beat this single-model bound.

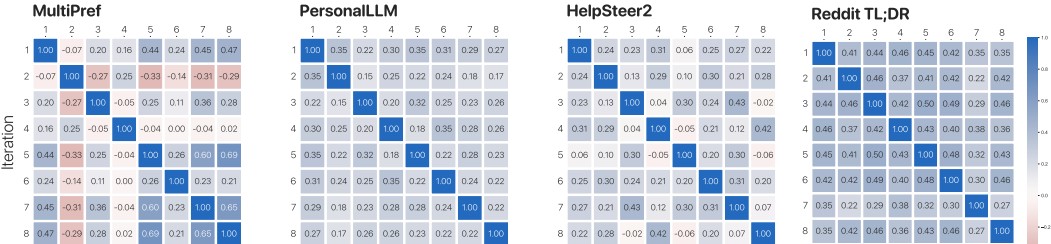

Figure 3: Pairwise Kendall–$\tau$ correlation scores between reward models in each ensemble, obtained by ranking 100 high-temperature continuations for 50 prompts. A score of 0 corresponds to random rankings, while a score of 1 corresponds to identical rankings; lower scores thus indicate greater diversity. The REDDIT TL;DR dataset—the smallest and focused on summarization—records the highest scores.

Thus, combining a handful of weaker reward models delivers a calibration accuracy that no single deterministic model can match.

Note that performance is reported as mean-squared calibration error. Other empirical pluralistic alignment experiments frequently report *accuracy*, but this is not meaningful here because the task is not majority classification and annotator identities are unavailable (see Appendix E).

**Do the reward models capture distinct preferences?** To test diversity at the policy level, we use Best-of-$N$ (BoN) sampling as a proxy for fine-tuning [64, 65]. Concretely, we select 50 prompts that the ensemble was not trained on, generate 100 diverse continuations from the frozen SFT model using high-temperature sampling, and score each response with every reward model in the ensemble. We quantify distinctness by the normalized Kendall–$\tau$ rank correlation coefficient [66], which counts the number of pairwise disagreements between two ranking lists, averaged over all prompts. Prompts for the first three datasets were taken from the PRISM dataset [8] (covering value-centric and controversial topics); prompts for the Reddit dataset were taken from its validation set.

The reward models within each ensemble are distinct and qualitatively different (Figure 3). This diversity is further illustrated by comparing the top-ranked response chosen by each model (Appendix G).

## 6 Discussion

Our approach raises several important considerations regarding its scope, alternatives, and limitations.

First, for highly contested or sensitive issues, pairwise calibration may not always be appropriate. We are not aiming for the model to express its *own* opinion or mirror the population; rather, a neutral answer—or, when necessary, outright refusal—is often the safer choice. However, while refusal policy is a simple fallback for disallowed or highly sensitive content, over-reliance can hamper the system's usefulness when thoughtful, context-aware answers are needed. Even ostensibly neutral responses may embed hidden framing biases, so it remains essential to understand and address those biases and to let the model meaningfully adapt to different user contexts, cultures, and values whenever such adaptation is appropriate.

Second, one might ask whether a single, unified model could achieve similar pluralistic alignment via alternate techniques. Indeed, one possible approach is to directly instruct the model within its context to adopt a particular perspective—known as *in-context steering* or *persona modulation* [27, 28, 67, 68]. However, recent empirical evidence suggests that language models instructed to emulate specific viewpoints frequently produce inconsistent or overly stereotyped responses, and often fail to capture nuanced differences between distinct user groups [69]. By contrast, our approach explicitly incorporates architectural support for pluralism by training multiple distinct reward functions.

Finally, while pairwise calibration ensures that the ensemble accurately reflects the preferences of annotators for *pairs* of responses given a context, it does not guarantee accurate representation of higher-order judgments over larger sets of responses. Ideally, for any reasonably sized set of $t$ responses, the proportion of reward functions ranking a given response highest would (approximately)

match this preference in the population. However, recovering such higher-order preference structures is information-theoretically infeasible using pairwise data alone. Capturing these relationships would require richer elicitation, such as rankings or best-of-$\ell$ judgments over larger response sets.

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

## A  Policy-Level Calibration

Here we relate pairwise calibration of reward functions to calibration at the policy level and motivate our binary indicator definition.

Recall in Definition 1 that an ensemble $(r_{\theta_j}; \alpha_j)_{j \in [k]}$ is perfectly pairwise calibrated when, over $x \sim D_{\mathcal{X}}, y_1, y_2 \sim \pi_0(\cdot \mid x)$, almost surely

$$p^\star(x, y_1, y_2) = \hat{p}(x, y_1, y_2) = \sum_{j=1}^{k} \alpha_j \, \mathbf{1} \left[ r_{\theta_j}(x, y_1) > r_{\theta_j}(x, y_2) \right].$$

The right-hand side represents the likelihood that a random reward function sampled from the mixture prefers $y_1 \succ y_2 \mid x$. Intuitively, we would like to say that the same holds over the learned mixture over policies $(\pi_j; \alpha_j)_{j \in [k]}$. This definition implicitly assumes that the policy $\pi_j$ learned from the $j$-th reward function $r_{\theta_j}$, conditioned on choosing between $y_1$ and $y_2$ for a given context $x$, always prefers the one with the higher reward. That is, if $r_{\theta_j}(x, y_1) > r_{\theta_j}(x, y_2)$, then $\Pr_{y \sim \pi_j(\cdot \mid x)}[y = y_1 \mid y \in \{y_1, y_2\}] \approx 1$.

In practice, however, this assumption does not strictly hold. During the fine-tuning, policies $\pi$ are typically learned by maximizing a regularized reward objective of the form:

$$\mathbb{E}_{x \sim D, \, y \sim \pi(\cdot \mid x)} \left[ r_\theta(x, y) \right] - \beta D_{\mathrm{KL}} \big( \pi(y \mid x) \| \pi_{\mathrm{ref}}(y \mid x) \big),$$

where $\beta$ controls the deviation of the fine-tuned policy from a reference policy $\pi_{\mathrm{ref}}$. The optimal policy under this objective, $\pi^\star$, is known to take the softmax form [70]:

$$\pi^\star(y \mid x) \propto \pi_{\mathrm{ref}}(y \mid x) \exp \left( \frac{1}{\beta} r_\theta(x, y) \right).$$

For a given triple $(x, y_1, y_2)$, this implies:

$$\Pr_{y \sim \pi^\star(\cdot \mid x)}[y = y_1 \mid y \in \{y_1, y_2\}]$$

$$= \frac{\pi_{\mathrm{ref}}(y_1 \mid x) \exp \left( \frac{1}{\beta} r_\theta(x, y_1) \right)}{\pi_{\mathrm{ref}}(y_1 \mid x) \exp \left( \frac{1}{\beta} r_\theta(x, y_1) \right) + \pi_{\mathrm{ref}}(y_2 \mid x) \exp \left( \frac{1}{\beta} r_\theta(x, y_2) \right)}$$

$$= \sigma \left( \frac{1}{\beta} \big( r_\theta(x, y_1) - r_\theta(x, y_2) \big) + \log \frac{\pi_{\mathrm{ref}}(y_1 \mid x)}{\pi_{\mathrm{ref}}(y_2 \mid x)} \right),$$

where $\sigma$ is the sigmoid function.

Let us denote this quantity by $\bar{p}_\theta(x, y_1, y_2)$. A natural alternative definition to perfect calibration would be to substitute $\mathbf{1}[r_\theta(x, y_1) > r_\theta(x, y_2)]$ with $\bar{p}$, i.e., would then use the formula:

$$\hat{p}(x, y_1, y_2) = \sum_{j=1}^{k} \alpha_j \cdot \bar{p}_{\theta_j}(x, y_1, y_2),$$

which more directly captures the probability that the mixture prefers $y_1$ over $y_2$ given $x$. However, this definition introduces additional complexity—it entangles the calibration objective with the reference policies, making both theoretical analysis and optimization more cumbersome.

We therefore abstract away these complications and keep the binary formulation as a tractable proxy for the more realistic notion. Notably, as $\beta \to 0$, the softmax behavior becomes increasingly sharp, and $\bar{p}_{\theta_j}(x, y_1, y_2) \to \mathbf{1}[r_{\theta_j}(x, y_1) > r_{\theta_j}(x, y_2)]$, as reward differences dominate. Thus, our binary definition can be viewed as capturing the low-regularization regime while remaining analytically and computationally manageable.

## B  Related Work

A growing set of proposals advocate for *pluralistic alignment*—modeling multiple perspectives in parallel so as to better capture the breadth of human judgments [23, 22, 9].

**Scalar Reward Aggregation Approaches.** To mitigate the majority-bias of a single reward, recent methods introduce *additional* loss terms or algorithmic modifications that slow or penalize the policy from over-optimizing for the majority. For instance, Xiao et al. [16] add a custom regularizer to reduce preference collapse in PPO updates, while Chen et al. [33] temper the update magnitude whenever annotator feedback shows high dispersion. Beyond such direct regularization, some works increase the reward model's capacity to represent preference diversity—for example, by weighting pairwise data based on annotator or context uncertainty [34] or learning a Bayesian distribution over reward function parameters [35]. Despite these refinements, methods reliant on a single scalar reward still risk oversimplifying genuine diversity in user opinions, motivating richer multi-reward solutions.

**Clustering and Group-Based Reward Modeling.** A more direct way to represent divergent preferences is to train multiple reward models, each matched to a distinct segment of the human population [9, 24–26]. This can be accomplished by clustering annotators based on demographics or inferred preference patterns and then fitting a group-specific reward function. For example, Chakraborty et al. [9] show that optimizing a mixture of cluster-specific rewards via a max-min objective can protect minority viewpoints from being overshadowed by the majority. Although powerful, these methods typically rely on stable, coherent groups, which may not reflect real-world fluidity in user preferences [27], and group membership can sometimes be ambiguous or proprietary. Additionally, overly granular clusters risk sparse data, whereas coarse groupings might obscure meaningful sub-group heterogeneity [28]. By contrast, we avoid the need for explicit clusters or demographic tags. Our approach learns a small ensemble of reward functions—each representing an internally consistent viewpoint—and ensures pairwise calibration so that overall, the ensemble reflects the full distribution of observed human judgments.

**Personalized and Identity-Conditional Alignment.** A further alternative is to directly personalize the alignment objective at the level of individual annotators, assuming each user has a unique reward function [36]. While highly expressive, such methods become cumbersome at scale, and excessive personalization can reinforce biases or amplify social divides [13]. Consequently, many personalization approaches use latent embeddings or partial fine-tuning, striking a balance between capturing user-specific signals and avoiding an explosion of fully individualized models. In practice, personalization-based methods still risk "echo chambers," prompting calls for a more bounded approach to preference diversity [13]. We share this concern, and thus focus on small ensembles as a middle path: each ensemble component aligns with a coherent subset of preferences, preserving pluralism while limiting the societal risks of extreme personalization.

## C  Additional Discussion Surrounding Theorem 1

### C.1  Ensemble Size Bound for Perfect Calibration with Finite Support

Let $y_1, \ldots, y_m$ be the elements of $\mathcal{Y}$. For a reward function $\theta \in \Theta$, let $\mathbf{x}^\theta \in \{0,1\}^{\binom{m}{2}}$ denote its *incidence vector*. We index $\mathbf{x}^\theta$ by pairs $ij$ with $i < j$, where $x^\theta_{ij} = \mathbf{1}[r_\theta(y_i) \geq r_\theta(y_j)]$. Observe that, for each $i \in \mathcal{N}$, their reward vector corresponds to some incidence vector. Thus, the vector $\mathbf{x}$ with $x_{ij} = p^\star(x, y_i, y_j)$ must live in the convex hull of incidence vectors; in particular, it is the convex combination of incidence vectors placing $p_\theta$ on $\mathbf{x}^\theta$ where $p_\theta$ is the fraction of $\mathcal{N}$ that have a reward function $r_\theta$. Since $\mathbf{x} \in \mathbb{R}^{\binom{m}{2}}$, Carathéodory's theorem [71] guarantees that it can be represented by a convex combination of only $\binom{m}{2} + 1$ vertices of the hull. This induces a $(\binom{m}{2} + 1)$-ensemble that is perfectly pairwise calibrated.

### C.2  Connections with Social Choice Theory

The problem we study can be expressed as a social-choice problem when $|\mathcal{X}| = 1$ and $|\mathcal{Y}| = m$. Each response plays the role of a candidate, and each annotator a voter who ranks these $m$ candidates. The *tournament graph* is the complete directed graph on these $m$ nodes, with an edge $(a, b)$ weighted by the fraction of voters who prefer $a$ to $b$. Under this reformulation, finding a perfectly pairwise calibrated ensemble is equivalent to finding a distribution over rankings that induces a given tournament graph. While an application of Carathéodory's theorem implies that a polynomial support size in the number of distinct pairs is sufficient ($\binom{m}{2} + 1$ in this context), we prove that it is NP-hard to determine whether such a distribution over rankings exists for a given

weighted directed graph. Consequently, even deciding whether a tournament graph is representable as any mixture of linear orders is itself NP-hard.

# D  Deferred Proofs

## D.1  Proof of Complexity (Theorem 1)

For convenience, we adopt standard terminology of *social choice theory* [42] in this proof. Specifically, we call elements of $\mathcal{Y} = \{y_1, \ldots, y_m\}$ *candidates*. Let $\mathrm{LO}(\mathcal{Y})$ denote the set of linear orders (or rankings) over $\mathcal{Y}$. The preferences of the voters (i.e., annotators) are represented by a distribution over rankings. A distribution assigns a probability $p_\sigma \geq 0$ to each $\sigma \in \mathrm{LO}(\mathcal{Y})$ such that $\sum_{\sigma \in \mathrm{LO}(\mathcal{Y})} p_\sigma = 1$.

Each ranking $\sigma$ can be represented by its *incidence vector* $\mathbf{x}^\sigma \in \{0,1\}^{\binom{m}{2}}$. For each pair $\{y_i, y_j\}$ with $i < j$,

$$x_{ij}^\sigma = \begin{cases} 1 & \text{if } y_i \text{ is ranked before } y_j \text{ in } \sigma, \\ 0 & \text{if } y_j \text{ is ranked before } y_i \text{ in } \sigma. \end{cases}$$

A *(weighted) tournament graph* corresponding to a distribution over rankings is the weighted average of these incidence vectors $\mathbf{t} = \sum_{\sigma \in \mathrm{LO}(\mathcal{Y})} p_\sigma \mathbf{x}^\sigma$. Each coordinate $t_{ij}$ represents the fraction of voters that rank $y_i$ before $y_j$. The set of all realizable tournament graphs is the convex hull of the incidence vectors:

$$\mathrm{TG} = \mathrm{conv}\left\{ \mathbf{x}^\sigma \mid \sigma \text{ is a linear ordering} \right\}.^6$$

We show that deciding if a given vector $\mathbf{y} \in \mathbb{R}^{\binom{m}{2}}$ is a valid tournament graph (i.e., $\mathbf{y} \in \mathrm{TG}$) is NP-hard. This implies that the potentially harder problem of finding a distribution over rankings $p_\sigma$ consistent with a given $\mathbf{t} \in TG$ is also NP-hard.

To establish this, we reduce from the NP-hard problem of *Minimum Feedback Arc Set (MFAS)* [72]. Given a directed graph $G = (V, E)$ on $m$ vertices (which we identify with $\mathcal{Y}$) and an integer $k$, MFAS asks if there is an ordering of the vertices $\sigma$ such that at most $k$ edges in $E$ are *feedback arcs* (i.e., an edge $(y, y')$ such that $y'$ appears before $y$ in $\sigma$). The number of feedback arcs for an ordering $\sigma$ is given by the linear function

$$N_{\mathrm{FAS}}(x^\sigma) = \sum_{\substack{i < j \ (y_j, y_i) \in E}} x_{ij}^\sigma + \sum_{\substack{i < j \ (y_i, y_j) \in E}} (1 - x_{ij}^\sigma).$$

Furthermore, since $N_{\mathrm{FAS}}$ is a linear function, its minimum on the convex hull is achieved at a vertex. Hence, $\min_{\sigma \in \mathrm{LO}(\mathcal{Y})} N_{\mathrm{FAS}}(\mathbf{x}^\sigma) = \min_{\mathbf{x} \in TG} N_{\mathrm{FAS}}(\mathbf{x})$.

Suppose we have a polynomial-time algorithm (an oracle) that decides whether a given (rational) vector $\mathbf{y}$ is in TG (in particular, this algorithm should run in time polynomial in the binary encoding of $\mathbf{y}$). We call this the *Tournament-Oracle Problem*. We show this allows for a polynomial-time algorithm for MFAS using the following theorem:

**Theorem 5** (Grötschel et al. [73], Corollary 4.3.12 (rephrased)). *Let $K \subset \mathbb{R}^d$ be a convex set, such that there exists a point $\mathbf{x}_0 \in \mathbb{R}^d$, and numbers $0 < r < R$ such that $B(\mathbf{x}_0, r) \subseteq K \subseteq B(\mathbf{x}_0, R)$, where $B(\mathbf{x}_0, r)$ is the ball of radius $r$ centered at $\mathbf{x}_0$. Suppose we have a membership oracle $\mathcal{O}$ for $K$. Then, for every $\mathbf{c} \in \mathbb{R}^d$, there exists an algorithm that outputs a point $\mathbf{z} \in \mathbb{R}^d$ such that*

$$\mathbf{c}^\top \mathbf{z} \leq \min_{\mathbf{x} \in K} \mathbf{c}^\top \mathbf{x} + \varepsilon \left( \max_{\mathbf{x} \in K} \mathbf{c}^\top \mathbf{x} - \min_{\mathbf{x} \in K} \mathbf{c}^\top \mathbf{x} \right).$$

*The algorithm runs in time polynomial in $\log(1/\varepsilon)$, $\log(R/r)$, the bit-size of $\mathbf{c}$, and $d$. The membership oracle is queried with rational points of polynomially bounded encoding sizes.*

We apply Theorem 5 with $K = \mathrm{TG}$, hence $d = \binom{m}{2} = O(m^2)$. Let $\mathbf{x}_0 \in \mathbb{R}^d$ be the point with all coordinates $1/2$. Since $\mathbf{x}_0$ can be induced by the average of all $m!$ possible rankings $\mathbf{x}^\sigma$, it is in TG. The squared $L_2$ distance from $\mathbf{x}_0$ to any vertex $\mathbf{x}^\sigma \in \{0,1\}^d$ is

$$\|\mathbf{x}^\sigma - \mathbf{x}_0\|_2^2 = \sum_{k=1}^d \left( \mathbf{x}_k^\sigma - \frac{1}{2} \right)^2 = \frac{d}{4},$$

---

[6]This is often called the *linear ordering polytope*.

implying that $R \le \sqrt{d/2} = O(m)$. McGarvey's theorem [74] shows that there exist tournaments with 0 or 1 in a single coordinate and $1/2$ everywhere else. Convex combinations of McGarvey's tournament graphs and $\mathbf{x}_0$ can generate any tournament such that every coordinate is within $1/\binom{m}{2}$ of $1/2$. Hence, $r = 1/\sqrt{\binom{m}{2}} \ge 1/m$ is sufficient. Thus, $\log(R/r) = O(\log m)$.

For MFAS, we want to minimize $N_{\text{FAS}}$. The objective function's linear part is defined by $\mathbf{c}_{\text{FAS}}$, whose components are in $\{-1, 0, 1\}$, and therefore $O(d) = O(m^2)$ bit complexity. The minimum value of $N_{\text{FAS}}$, as it must occur at a vertex in $\{0, 1\}^d$, is an integer. Furthermore, $0 \le N_{\text{FAS}}(\mathbf{x}) \le \binom{m}{2}$ for all $\mathbf{x} \in \text{TG}$.

Applying the theorem with $\varepsilon < 1/(2 \cdot \binom{m}{2})$ and rounding the solution gives a polynomial time algorithm for finding the optimal value of MFAS, which, assuming $P \ne NP$, is a contradiction. $\square$

## D.2 Proof of Approximation Guarantee (Theorem 2)

Fix $\varepsilon > 0$. For each $i \in \mathcal{N}$, let $r_{\theta_i}$ be the reward function that induces $i$'s preferences (i.e., $y_1 \succ y_2 \mid x \iff r_{\theta_i}(x, y_1) > r_{\theta_i}(x, y_2)$). Let $k = \lceil \frac{1}{4\varepsilon} \rceil$. This choice implies $k \ge \frac{1}{4\varepsilon}$, so $\varepsilon \le \frac{1}{4k}$.

Consider randomly constructing a $k$-ensemble $\mathbf{r}$ as follows: Randomly select $k$ annotators $i_1, \ldots, i_k \sim \mathcal{N}$ uniformly with replacement. The ensemble is then:

$$\mathbf{r} = (r_{\theta_{i_1}}, \ldots, r_{\theta_{i_k}}; 1/k, \ldots, 1/k).$$

Let $\mathcal{R}$ denote this distribution over ensembles.

For each triple $(x, y_1, y_2)$ consider

$$\mathbb{E}_{\mathbf{r} \sim \mathcal{R}} \left[ (\hat{p}^{\mathbf{r}}(x, y_1, y_2) - p^*(x, y_1, y_2))^2 \right].$$

The estimator $\hat{p}^{\mathbf{r}}(x, y_1, y_2)$ is obtained by drawing

$$T \sim \text{Bin}(k, p^*(x, y_1, y_2)),$$

and setting $\hat{p}^{\mathbf{r}}(x, y_1, y_2) = T/k$. Therefore

$$\mathbb{E}_{\mathbf{r} \sim \mathcal{R}} \left[ (\hat{p}^{\mathbf{r}}(x, y_1, y_2) - p^*(x, y_1, y_2))^2 \right] = \frac{\text{Var}(T)}{k^2} \le \frac{p^*(x, y_1, y_2)(1 - p^*(x, y_1, y_2))}{k} \le \frac{1}{4k}.$$

Next, the expected calibration error of a randomly chosen ensemble $\mathbf{r}$ satisfies

$$\mathbb{E}_{\mathbf{r} \sim \mathcal{R}} \left[ \mathbb{E}_{x \sim D_{\mathcal{X}}, y_1, y_2 \sim \pi_0(\cdot \mid x)} \left( \hat{p}^{\mathbf{r}}(x, y_1, y_2) - p^*(x, y_1, y_2) \right)^2 \right]$$

$$= \mathbb{E}_{x \sim D_{\mathcal{X}}, y_1, y_2 \sim \pi_0(\cdot \mid x)} \left[ \mathbb{E}_{\mathbf{r} \sim \mathcal{R}} \left( \hat{p}^{\mathbf{r}}(x, y_1, y_2) - p^*(x, y_1, y_2) \right)^2 \right]$$

$$\le \mathbb{E}_{x \sim D_{\mathcal{X}}, y_1, y_2 \sim \pi_0(\cdot \mid x)} \left[ \frac{1}{4k} \right] = \frac{1}{4k}.$$

Since the expected calibration error of a randomly chosen $\mathbf{r} \sim \mathcal{R}$ is at most $\frac{1}{4k}$, this implies that the same bound holds for at least one deterministic $\mathbf{r}$, as needed. $\square$

## D.3 Proof of Outlier Bound (Theorem 3)

Let $\mathbf{r} = (r_{\theta_j}; \alpha_j)_{j=1}^k$ be a $k$-ensemble that is $\varepsilon$-pairwise calibrated. Fix $\beta \ge 2$, and let $\gamma = (\beta + 1)\sqrt{\varepsilon}$. Recall the disagreement score for a reward function $r_\theta$:

$$\Phi(\theta) = \Pr_{\substack{x \sim D_{\mathcal{X}}, y_1, y_2 \sim \pi_0(\cdot \mid x), \\ i \sim \mathcal{N}}} \left[ \mathbf{1}[r_\theta(x, y_1) > r_\theta(x, y_2)] \ne \mathbf{1}[y_1 \succ_i y_2 \mid x] \right].$$

Let $\Phi_{\min} = \inf_\theta \Phi(\theta)$. Then, $r_\theta$ is a $(\beta, \gamma)$-outlier if $\Phi(\theta) > \beta \cdot \Phi_{\min} + \gamma$.

Define the random variables

$$V^\theta = \mathbf{1}\{r_\theta(x, y_1) > r_\theta(x, y_2)\}, \qquad P^\star = p^\star(x, y_1, y_2), \qquad \hat{P} = \hat{p}(x, y_1, y_2),$$

The randomness for these variables is over the draw of a triple $(x, y_1, y_2)$ according to the underlying distribution (e.g., $x \sim D_{\mathcal{X}}, y_1, y_2 \sim \pi_0(\cdot \mid x)$). For any two random variables $X$ and $Y$ (over triples), let

$$d(X, Y) = \mathbb{E}_{x \sim D_{\mathcal{X}}, y_1, y_2 \sim \pi_0(\cdot \mid x)}[|X - Y|].$$

First, we show that $\Phi(\theta) = d(V^\theta, P^\star)$. To this end, we can rewrite the disagreement score as

$$\Pr_{\substack{x \sim D_{\mathcal{X}}, y_1, y_2 \sim \pi_0(\cdot \mid x), \\ i \sim \mathcal{N}}} [\mathbf{1}[r_\theta(x, y_1) > r_\theta(x, y_2)] \neq \mathbf{1}[y_1 \succ_i y_2 \mid x]]$$

$$= \mathbb{E}_{x \sim D_{\mathcal{X}}, y_1, y_2 \sim \pi_0(\cdot \mid x)} \left[ \Pr_{i \sim \mathcal{N}}[\mathbf{1}[r_\theta(x, y_1) > r_\theta(x, y_2)] \neq \mathbf{1}[y_1 \succ_i y_2 \mid x]] \right].$$

Furthermore, for a fixed triple $(x, y_1, y_2)$, the inner term is:

$$\Pr_{i \sim \mathcal{N}}[\mathbf{1}[r_\theta(x, y_1) > r_\theta(x, y_2)] \neq \mathbf{1}[y_1 \succ_i y_2 \mid x]]$$

$$= V^\theta \cdot \left(1 - \Pr_{i \sim \mathcal{N}}[y_1 \succ y_2 \mid x]\right) + (1 - V^\theta) \cdot \Pr_{i \sim \mathcal{N}}[y_1 \succ y_2 \mid x]$$

$$= V^\theta \cdot (1 - P^\star) + (1 - V^\theta) \cdot P^\star.$$

Since $V^\theta$ is binary and $P^\star \in [0, 1]$, this simplifies to $|V^\theta - P^\star|$. Thus, the disagreement score is indeed $\Phi(\theta) = d(V^\theta, P^\star)$.

Define $\hat{\Phi}(\theta) = d(V^\theta, \hat{P})$ and $\hat{\Phi}_{\min} = \inf_\theta \hat{\Phi}(\theta)$ as an analogous "disagreement score" with respect to $\hat{P}$. Next, we claim that if $\theta$ is a $(\beta, \gamma)$-outlier, then $d(V^\theta, \hat{P}) > \beta \cdot \hat{\Phi}_{\min}$. By the triangle inequality,

$$|d(V^\theta, P^\star) - d(V^\theta, \hat{P})| \leq d(P^\star, \hat{P}).$$

The ensemble $\mathbf{r}$ is $\varepsilon$-pairwise calibrated, meaning $\mathbb{E}[(\hat{P} - P^\star)^2] \leq \varepsilon$. By Jensen's inequality,

$$\mathbb{E}[|\hat{P} - P^\star|]^2 \leq \mathbb{E}[(\hat{P} - P^\star)^2] \leq \varepsilon.$$

Hence, $d(P^\star, \hat{P}) \leq \sqrt{\varepsilon}$. So, for all $\theta$, we have

$$|d(V^\theta, \hat{P}) - d(V^\theta, P^\star)| \leq \sqrt{\varepsilon}.$$

This also implies that $|\Phi_{\min} - \hat{\Phi}_{\min}| \leq \sqrt{\varepsilon}$.

Combining these observations, we see that if $\theta$ is a $(\beta, \gamma)$-outlier, then $d(V^\theta, P^\star) > \beta \cdot \Phi_{\min} + (\beta + 1) \cdot \sqrt{\varepsilon}$, which implies that $d(V^\theta, \hat{P}) > \beta \hat{\Phi}_{\min}$.

Next, we claim that for a fixed $\theta \in \Theta$,

$$d(V^\theta, \hat{P}) = \mathbb{E}_{\theta' \sim \mathbf{r}}[d(V^\theta, V^{\theta'})],$$

where $\theta' \sim \mathbf{r}$ denotes drawing from the ensemble, i.e., $\theta_j$ with probability $\alpha_j$. Expanding the definitions:

$$\mathbb{E}_{x \sim D_{\mathcal{X}}, y_1, y_2 \sim \pi_0(\cdot \mid x)}[|V^\theta - \hat{P}|] = \mathbb{E}_{\theta' \sim \mathbf{r}}[\mathbb{E}_{x \sim D_{\mathcal{X}}, y_1, y_2 \sim \pi_0(\cdot \mid x)}[|V^\theta - V^{\theta'}|]].$$

Swapping the order of the expectations:

$$\mathbb{E}_{x \sim D_{\mathcal{X}}, y_1, y_2 \sim \pi_0(\cdot \mid x)}[|V^\theta - \hat{P}|] = \mathbb{E}_{x \sim D_{\mathcal{X}}, y_1, y_2 \sim \pi_0(\cdot \mid x)}[\mathbb{E}_{\theta' \sim \mathbf{r}}[|V^\theta - V^{\theta'}|]].$$

For a fixed triple $(x, y_1, y_2)$, the equality $|V^\theta(x, y_1, y_2) - \hat{P}(x, y_1, y_2)| = \mathbb{E}_{\theta' \sim \mathbf{r}}[|V^\theta(x, y_1, y_2) - V^{\theta'}(x, y_1, y_2)|]$ holds, because $V^\theta(x, y_1, y_2)$ is deterministic (in $\{0, 1\}$, and $V^{\theta'}$ is a Bernoulli random variable whose expectation is $\hat{p}(x, y_1, y_2)$.

By Markov's inequality, for any $\theta$, we have that

$$\Pr_{\theta' \sim \mathbf{r}}[d(V^{\theta'}, V^\theta) \geq (\beta - 1) \cdot d(V^\theta, \hat{P})]$$

$$= \Pr_{\theta' \sim \mathbf{r}}[d(V^{\theta'}, V^\theta) \geq (\beta - 1) \cdot \mathbb{E}_{\theta' \sim \mathbf{r}}[d(V^{\theta'}, V^\theta)]] \leq \frac{1}{\beta - 1}.$$

By the triangle inequality, $d(V^{\theta'}, \hat{P}) \leq d(V^{\theta'}, V^{\theta}) + d(V^{\theta}, \hat{P})$. Therefore, if $d(V^{\theta'}, \hat{P}) \geq \beta \cdot d(V^{\theta}, \hat{P})$, that implies that $d(V^{\theta'}, \hat{P}) \geq (\beta - 1) \cdot d(V^{\theta}, V^{\theta'})$. This gives us that

$$\Pr_{\theta' \sim \mathbf{r}}[d(V^{\theta'}, \hat{P}) \geq \beta \cdot d(V^{\theta}, \hat{P})] \leq \frac{1}{\beta - 1}.$$

Taking the infimum over choices of $\theta$ yields (e.g., using $\theta$ such that $d(V^{\theta}, \hat{P}) \to \hat{\Phi}_{\min}$)

$$\Pr_{\theta' \sim \mathbf{r}}[d(V^{\theta'}, \hat{P}) > \beta \cdot \hat{\Phi}_{\min}] \leq \frac{1}{\beta - 1}.$$

Finally, as $(\beta, \gamma)$-outliers must satisfy $d(V^{\theta'}, \hat{P}) > \beta \cdot \hat{\Phi}_{\min}$, this immediately implies

$$\Pr_{\theta' \sim \mathbf{r}}[\theta' \text{ is a } (\beta, \gamma)\text{-outlier}] \leq \frac{1}{\beta - 1},$$

yielding the first part of the theorem.

Now, construct $\mathbf{r}'$ as follows: Order the $\theta_1, \ldots, \theta_k$ from $\mathbf{r}$ in non-decreasing order by $d(\theta, \hat{P})$ as $\theta_{i_1}, \ldots, \theta_{i_k}$. Remove $\theta_j$ if they are in the top $\frac{1}{\beta - 1}$ of total weight according to this ordering (i.e., if $j$ satisfies that $\sum_{j'=j}^{k} \alpha_{j'} \leq \frac{1}{\beta - 1}$). Then, renormalize the remaining $\alpha_j$s to sum to 1. The previously derived bound shows that at most $\frac{1}{\beta - 1}$ of the probability mass is on $\theta$ satisfying $d(\theta, \hat{P}) > \beta \cdot \Phi_{\min}$. Removing those with largest $d(\cdot, \hat{P})$ values ensures all such $\theta$ are removed. By the above arguments, this also implies that $\mathbf{r}'$ contains no $(\beta, \gamma)$-outliers. Furthermore, this computation depended only on $\mathbf{r}$ (i.e., we did not need access to true proportions $p^{\star}$).

It remains to show that $\mathbf{r}'$ is $(\sqrt{\varepsilon} + \frac{1}{\beta - 1})^2$-pairwise calibrated. To this end, let $\hat{P}'$ be the analogous random variable to $\hat{P}$ for $\mathbf{r}'$, i.e., it takes on values $\hat{p}^{\mathbf{r}'}(x, y_1, y_2)$. We claim that $|\hat{P}' - \hat{P}| \leq \frac{1}{\beta - 1}$ surely.

Let $S_{\text{rem}}$ be the set of indices of removed $\theta_j$. Let $\alpha_{\text{rem}} = \sum_{j \in S_{\text{rem}}} \alpha_j$. We know $\alpha_{\text{rem}} \leq \frac{1}{\beta - 1}$. Let $\alpha_{\text{tot}} = \sum_{j \notin S_{\text{rem}}} \alpha_j = 1 - \alpha_{\text{rem}}$. For a fixed triple $(x, y_1, y_2)$, let $V_j = V^{\theta_j}(x, y_1, y_2) \in \{0, 1\}$. Then $\hat{P}(x, y_1, y_2) = \sum_{j \notin S_{\text{rem}}} \alpha_j V_j + \sum_{j \in S_{\text{rem}}} \alpha_j V_j$. And $\hat{P}'(x, y_1, y_2) = \frac{\sum_{j \notin S_{\text{rem}}} \alpha_j V_j}{\alpha_{\text{tot}}}$.

Consider the difference:

$$\hat{P}' - \hat{P} = \frac{\sum_{j \notin S_{\text{rem}}} \alpha_j V_j}{\alpha_{\text{tot}}} - \left( \sum_{j \notin S_{\text{rem}}} \alpha_j V_j + \sum_{j \in S_{\text{rem}}} \alpha_j V_j \right)$$

$$= \left( \frac{1}{\alpha_{\text{tot}}} - 1 \right) \sum_{j \notin S_{\text{rem}}} \alpha_j V_j - \sum_{j \in S_{\text{rem}}} \alpha_j V_j$$

$$= \frac{1 - \alpha_{\text{tot}}}{\alpha_{\text{tot}}} \sum_{j \notin S_{\text{rem}}} \alpha_j V_j - \sum_{j \in S_{\text{rem}}} \alpha_j V_j$$

$$= \frac{\alpha_{\text{rem}}}{\alpha_{\text{tot}}} \sum_{j \notin S_{\text{rem}}} \alpha_j V_j - \sum_{j \in S_{\text{rem}}} \alpha_j V_j.$$

Let $A = \sum_{j \notin S_{\text{rem}}} \alpha_j V_j$ and $B = \sum_{j \in S_{\text{rem}}} \alpha_j V_j$. We know $0 \leq A \leq \sum_{j \notin S_{\text{rem}}} \alpha_j = \alpha_{\text{tot}}$, and $0 \leq B \leq \sum_{j \in S_{\text{rem}}} \alpha_j = \alpha_{\text{rem}}$. So, $\hat{P}' - \hat{P} = \frac{\alpha_{\text{rem}}}{\alpha_{\text{tot}}} A - B$.

Now, $0 \leq A \leq \alpha_{\text{tot}}$ and $0 \leq B \leq \alpha_{\text{rem}}$, so this difference is bounded in $[-\alpha_{\text{rem}}, \alpha_{\text{rem}}]$ and thus bounded in $[-\frac{1}{\beta - 1}, \frac{1}{\beta - 1}]$.

Finally, we show the pairwise calibration bound. We have just shown that $|\hat{P} - \hat{P}'| \leq \frac{1}{\beta - 1}$, which implies that

$$\sqrt{\mathbb{E}_{x \sim D_{\mathcal{X}}, y_1, y_2 \sim \pi_0(\cdot|x)}[(\hat{P} - \hat{P}')^2]} \leq \frac{1}{\beta - 1}.$$

Furthermore, $\varepsilon$-pairwise calibration implies that

$$\sqrt{\mathbb{E}_{x \sim D_{\mathcal{X}}, y_1, y_2 \sim \pi_0(\cdot|x)}[(P^{\star} - \hat{P})^2]} \leq \sqrt{\varepsilon}.$$

By the triangle inequality for $L_2$ norms (Minkowski inequality),

$$\sqrt{\mathbb{E}_{x \sim D_{\mathcal{X}}, y_1, y_2 \sim \pi_0(\cdot | x)}[(P^\star - \hat{P}')^2]} \leq \sqrt{\varepsilon} + \frac{1}{\beta - 1}.$$

This implies that $\mathbf{r}'$ is $\left(\sqrt{\varepsilon} + \frac{1}{\beta-1}\right)^2$-pairwise calibrated, as needed. $\qquad\square$

## D.4 Proof of Biased Loss (Lemma 1)

First, note that by linearity over expectation,

$$\mathbb{E}_{\mathcal{D}}[\mathcal{L}(\mathbf{r})] = \mathbb{E}_{x \sim D_{\mathcal{X}}, y_1, y_2 \sim \pi_0(\cdot | x), p \sim \mathrm{Bin}(n, p^\star(x, y_1, y_2))/n}[(\hat{p}^{\mathbf{r}}(x, y_1, y_2) - p)^2].$$

Fix a triple $(x, y_1, y_2)$. For brevity, we will write $p^\star$ and $\hat{p}^{\mathbf{r}}$ instead of $p^\star(x, y_1, y_2)$ and $\hat{p}^{\mathbf{r}}(x, y_1, y_2)$, respectively. The inner term then becomes

$$\mathbb{E}_{p \sim \mathrm{Bin}(n, p^\star)/n}[(\hat{p}^{\mathbf{r}} - p)^2].$$

Furthermore, observe that $\mathbb{E}_{p \sim \mathrm{Bin}(n, p^\star)/n}[p] = p^\star$. Hence,

$$
\begin{aligned}
\mathbb{E}_{p \sim \mathrm{Bin}(n, p^\star)/n}[(\hat{p}^{\mathbf{r}} - p)^2] &= \mathbb{E}_{p \sim \mathrm{Bin}(n, p^\star)/n}[(\hat{p}^{\mathbf{r}} - p^\star + p^\star - p)^2] \\
&= \mathbb{E}_{p \sim \mathrm{Bin}(n, p^\star)/n}[(\hat{p}^{\mathbf{r}} - p^\star)^2 - 2(\hat{p}^{\mathbf{r}} - p^\star)(p^\star - p) + (p^\star - p)^2] \\
&= (\hat{p}^{\mathbf{r}} - p^\star)^2 - 2(\hat{p}^{\mathbf{r}} - p^\star)(p^\star - p^\star) + \mathbb{E}_{p \sim \mathrm{Bin}(n, p^\star)/n}[(p^\star - p)^2] \\
&= (\hat{p}^{\mathbf{r}} - p^\star)^2 + \mathbb{E}_{p \sim \mathrm{Bin}(n, p^\star)/n}[(p^\star - p)^2].
\end{aligned}
$$

Taking expectation over $x \sim D_{\mathcal{X}}, y_1, y_2 \sim \pi_0(\cdot | x)$ yields the lemma statement. $\qquad\square$

## D.5 Proof of Generalization Bound (Theorem 4)

We invoke the following uniform-convergence result:

**Theorem 6** (Mohri et al. [43], Theorem 11.8 (rephrased)). *Let $\mathcal{H}$ be a family of real-valued functions and*

$$\mathcal{G} = \big\{(x, y) \mapsto \mathcal{L}(h(x), y) \mid h \in \mathcal{H}\big\}$$

*the family of loss functions associated to $\mathcal{H}$. Assume that $\mathrm{Pdim}(\mathcal{G}) = d^\star$ and that the loss function $\mathcal{L}$ is non-negative and bounded by $M$. Let $\mathcal{D}$ be a distribution over $(x, y)$ and let $\bar{\mathcal{D}}$ be a sample of size $N$. Then, for any $\delta > 0$, with probability at least $1 - \delta$ over the choice of samples,*

$$\sup_{h \in \mathcal{H}} \Big| \mathbb{E}_{(x,y) \sim \mathcal{D}}[\mathcal{L}(h(x), y)] - \mathbb{E}_{(x,y) \sim \bar{\mathcal{D}}}[\mathcal{L}(h(x), y)] \Big| \leq M \sqrt{\frac{2d^\star \log(\frac{eN}{d})}{N}} + M \sqrt{\frac{\log \frac{1}{\delta}}{2N}}^{\,7}$$

For our purposes, $\mathcal{H}$ is the set of functions $\{\hat{p}^{\mathbf{r}} \mid \mathbf{r} \in \mathcal{R}_k\}$ where each $\hat{p}^{\mathbf{r}}$ maps $\mathcal{X} \times \mathcal{Y} \times \mathcal{Y} \to [0, 1]$. The loss function $\mathcal{L}$ is the squared error. The domain points are triples $(x, y_1, y_2)$, drawn from the usual $x \sim D_{\mathcal{X}}, y_1, y_2 \sim \pi_0(\cdot | x)$. The target the label is the empirical proportion

$$p \sim \mathrm{Bin}\big(n, p^\star(x, y_1, y_2)\big)/n,$$

Since $p \in [0, 1]$, $\mathcal{L}$ is bounded by $M = 1$.

Furthermore,

$$\hat{\mathcal{L}}(\mathbf{r}) = \mathbb{E}_{(x,y) \sim \bar{\mathcal{D}}}\big[\mathcal{L}(h(x), y)\big],$$

and by Lemma 1,

$$\mathcal{L}'(\mathbf{r}) = \mathbb{E}_{(x,y) \sim \mathcal{D}}\big[\mathcal{L}(h(x), y)\big].$$

It therefore suffices to show that $\mathrm{Pdim}(\mathcal{G}) \leq 20(d + 1)k \cdot \log(2(d + 1)k)$, where

$$\mathcal{G} = \{((x, y_1, y_2), p) \mapsto (\hat{p}^{\mathbf{r}}(x, y_1, y_2) - p)^2 \mid \mathbf{r} \in \mathcal{R}\}.$$

---

[7]The theorem statement in Mohri et al. [43] states one-sided error, but the proof implies two-sided one.

For notational convenience, inputs to function classes (e.g., $(x, y_1, y_2)$ or $((x, y_1, y_2), p)$) may be generically denoted by $z$.

Recall the following standard definitions and results from learning theory. Fix a function class $\mathcal{F}'$ and a set of $m$ points, $\{z_1, \ldots, z_m\}$. For a binary vector $b \in \{0,1\}^m$, called a *sign pattern*, $\mathcal{F}'$ realizes $b$ if there exists $f \in \mathcal{F}'$ such that $\mathbf{1}[f(z_i) \geq 0] = b_i$ for all $i$. We say $\mathcal{F}'$ *shatters* the set if all $2^m$ sign patterns are realized. The VC-dimension of $\mathcal{F}'$ is the size of the largest set it shatters. If $\mathcal{F}'$ is real-valued, its pseudo-dimension is the VC dimension of $\{(z, t) \mapsto f(z) - t \mid f \in \mathcal{F}'\}$. If the latter function class shatters a set, we say that $\mathcal{F}'$ pseudo-shatters that set. We write $\mathrm{Pdim}(\mathcal{F}')$ for the pseudo-dimension of $F'$. Finally, Sauer's Lemma [75] states that a class of VC-dimension $d'$ induces at most $(em/d')^{d'}$ sign patterns on $m$ points.

We upper-bound $\mathrm{Pdim}(\mathcal{G})$ by analyzing a sequence of function classes $\mathcal{F}_1, \mathcal{F}_2, \mathcal{F}_3$ and bounding their pseudo-dimensions $d_1, d_2, d_3$:

**1. Linear combinations:** Let $\mathcal{F}_1 = \{\sum_{j=1}^k \alpha_j \cdot f_j(z) \mid f_j \in \mathcal{F}, \alpha_j \in \mathbb{R}^k\}$, the set of $k$-sized linear combinations of functions in $\mathcal{F}$.

Fix $m$ points $(z_1, t_1), \ldots, (z_m, t_m)$. First, consider the maximum number of sign patterns of $z_1, \ldots, z_m$ which can be realized by (the binary-valued) $\mathcal{F}$. Since $\mathcal{F}$ has VC-dimension $d$, by Sauer's Lemma, this is at most $(e \cdot m/d)^d$. Thus, if we are picking $k$ such functions from $\mathcal{F}$, this can induce $(em/d)^{kd} \leq (em)^{kd}$ combinations of sign patterns.

Fixing a choice of patterns $\mathbf{v}_1, \ldots, \mathbf{v}_k \in \{0,1\}^m$. We now consider how many sign patterns of $\{(z_i, t_i)\}$ can be induced by various choices of $\boldsymbol{\alpha}$. Note that a certain $\boldsymbol{\alpha}$ will induce sign pattern with $b_i = \mathbf{1}[\boldsymbol{\alpha} \cdot (v_{1i}, \ldots, v_{ki}) - t_i \geq 0]$. Consider the set of $k$-dimensional linear functions mapping $\mathbb{R}^k \to \mathbb{R}$, $\{\mathbf{z} \mapsto \boldsymbol{\alpha} \cdot \mathbf{z} \mid \boldsymbol{\alpha} \in \mathbb{R}^k\}$. Note that this class has pseudo-dimension $k$ [43]. This immediately implies, by Sauer's Lemma and the definition of pseudo-dimension, that the number of sign patterns inducible on $\{(z_i, t_i)\}$ is at most $(em/k)^k \leq (em)^k$.

Together, these imply that the total number of sign patterns realizable by $\mathcal{F}_1$ is at most $(em)^{(d+1)k}$. Thus, if $\mathcal{F}_1$ can pseudo-shatter this set, all sign patterns must be realizable, so $2^m \leq (em)^{(d+1)k}$. Equivalently, $\log(2)m \leq (d+1)k \cdot \log(m)$, implying

$$m \leq \frac{(d+1)k}{\log(2)} \cdot \log(m) \leq 2 \cdot (d+1) \cdot k \cdot \log m.$$

By Lemma A.1 of Shalev-Shwartz and Ben-David [76], $m \leq 4(d+1)k \cdot \log(2(d+1)k)$. Hence, $d_1 \leq 4(d+1)k \cdot \log(2(d+1)k)$.

**2. Affine shifts:** Let $\mathcal{F}_2 = \{(z, y) \mapsto f(z) - y \mid f \in \mathcal{F}_1\}$.

If $\mathcal{F}_2$ pseudo-shatters

$$((z_1, y_1), t_1), \ldots, ((z_m, y_m), t_m),$$

then $\mathcal{F}_1$ pseudo-shatters

$$(z_1, t_1 - y_1), \ldots, (z_m, t_m - y_m),$$

implying that the pseudo-dimension of $\mathcal{F}_2$ is at most that of $\mathcal{F}_1$. Hence, $d_2 \leq d_1$.

**3. Squaring:** Let $\mathcal{F}_3 = \{z \mapsto f(z)^2 \mid f \in \mathcal{F}_2\}$.

Fix $m$ points $S = \{(z_1, t_1), \ldots (z_m, t_m)\}$ pseudo-shattered by $\mathcal{F}_3$. Assume each $t_i > 0$, as otherwise that point alone cannot be shattered. Furthermore assume that $f(z_i) \neq t_i$ for any $f \in \mathcal{F}_3$, otherwise we can adjust $t_i$ such that the set is still pseudo-shattered and equality does not hold.

Now, consider the sign patterns of function $f \in \mathcal{F}_2$ on the $2m$ points

$$S' = \{(z_1, \sqrt{t_1}), \ldots, (z_m, \sqrt{t_m}), (z_1, -\sqrt{t_1}), \ldots, (z_m, \sqrt{t_m})\}.$$

Observe that if two functions $f, f' \in \mathcal{F}_2$ have that $f(z)^2$ and $f'(z)^2$ differ in sign pattern on $S$, then $f(z)$ and $f'(z)$ differ in sign pattern on $S'$, as if $f(x_i)^2 < t_i$ and $f'(x_i)^2 \geq t_i$, then that implies $-\sqrt{t_i} < f(x_i) < \sqrt{t_i}$ and either $f'(x_i) > \sqrt{t_i}$ or $f'(x_i) < \sqrt{t_i}$, so $f$ and $f'$ must differ on at least one point. Therefore, the number of sign patterns $\mathcal{F}_2$ can realize on $S'$ must be at least as large as the number of sign patterns $\mathcal{F}_3$ can realize on $S$. By Sauer's lemma, the number of sign patterns $\mathcal{F}_2$

can realize on $S'$ is at most $\left(\frac{e \cdot 2m}{d_2}\right)^{d_2}$. Since $\mathcal{F}_3$ pseudo-shatters these points, $2^m \leq \left(\frac{e \cdot 2m}{d_2}\right)^{d_2}$, and, equivalently, $2^{m/d_2}/(m/d_2) \leq 2e$. Now if $m \geq 5d_2$, then, since $2^z/z$ is increasing for $z \geq 2$, the LHS is at least $2^5/5 > 2e$. Therefore, $d_3 \leq 5 \cdot d_2$.

Finally, observe that after all of these transformations, $\mathcal{G}$ is in fact a subset of $\mathcal{F}_3$. Therefore,

$$\mathrm{Pdim}(\mathcal{G}) \leq d_3 \leq 5d_2 \leq 5d_1 \leq 20(d+1)k\log\big(2(d+1)k\big),$$

as needed. $\square$

## E  BT Objective and Evaluation Metrics

As noted in Section 4, we replace the standard BCE objective with a soft-label MSE loss; below we detail this choice and the associated evaluation metrics.

**Soft-label objective.** The BT model assumes that human preferences arise from a single latent reward function $r^\star$ via

$$p^\star(y_1 \succ y_2 \mid x) = \sigma\big(r^\star(x, y_1) - r^\star(x, y_2)\big). \tag{3}$$

Maximum-likelihood estimation under this assumption yields the familiar BCE objective:

$$\mathcal{L}_{\mathrm{BCE}}(\theta) = -\mathbb{E}_{(x,y_1,y_2)\sim\mathcal{D}}\Big[\log \sigma\big(r_\theta(x, y_1) - r_\theta(x, y_2)\big)\Big].$$

Equation 3 views each comparison as a noisy glimpse of one ground-truth, $r^\star$, treating annotators as interchangeable sensors of a single value system. Our approach flips that premise: the goal is to reproduce the observed preference fractions rather than denoise them. We therefore retain the fractional targets and train using

$$\mathcal{L}_{\mathrm{Soft\ Labels}}(\theta) = \mathbb{E}_{(x,y_1,y_2)\sim\mathcal{D}}\big[\ell\big(p(x, y_1, y_2), \hat{p}_\theta(x, y_1, y_2)\big)\big],$$

where $\ell$ is any proper loss. Here $\theta$ indexes a single base reward model.

**Evaluation metrics.** After BCE training, accuracy—i.e., how often the model's decision agrees with the majority—is the standard metric. In this work, our objective is not to predict the single majority decision nor to predict the preference decision of an underlying group of annotators. Thus, Accuracy is ill-suited. Instead, *for the ensemble*, we report the mean-squared calibration error, also referred to as the *Brier score*, which directly measures how closely the model's predicted probabilities align with the observed preference fractions.

For a single reward, since the output is taken to be binary, the relevant metric is regret, defined as

$$\mathrm{Regret} = \mathbf{1}\{\hat{p} > 0.5\}(1 - p) + \mathbf{1}\{\hat{p} \leq 0.5\}p - \min\{p, 1 - p\},$$

which measures the extra error incurred over the Bayes-optimal choice.

|        |      | Prediction | |
|--------|------|------------|------|
|        |      | Hard | Soft |
| **Labels** | Hard | Accuracy | Binary Cross-Entropy |
|        | Soft | Regret | Brier Score |

## F  Experiments

### F.1  Training Pipeline

**Optimization.** We fine-tune all model parameters, including both the base transformer and the final linear reward head, using the SOAP optimizer [77], which we found to accelerate training compared to AdamW. Given the relatively small size of the preference datasets compared to the model's capacity, we train for only a single epoch, generally sufficient to achieve convergence without overfitting, as demonstrated by previous reward-model training studies [62, 64, 54, 78, 79, 53, 56]. Training is conducted with BF16 precision on a single NVIDIA H100 GPU, utilizing gradient accumulation to accommodate an effective batch size of up to 512. We adopt learning rates in the range $\{1e{-}5 \ldots 5e{-}5\}$ with a cosine decay schedule, a linear warmup spanning the first $3\%$ of training steps, and weight decay set to $0.1$.

| Name | Pref. Pairs | Prompts (Train/Test) | Prompt Source | Response Source | Annotation Method | Avg # Annots. |
|------|-------------|----------------------|---------------|-----------------|-------------------|---------------|
| MultiPref | 9,413 | 4,791 / 532 | Anthropic HH, ChatArena, ShareGPT | GPT-4, GPT-3.5, Llama-2/3, Tulu-2 | Human | 4.00 |
| PersonalLLM | 263,256 | 9,402 / 1,000 | RewardBench, Anthropic HH, Help-Steer | GPT-4, Claude-3, Llama-3, Gemini-Pro | AI | 10 |
| HelpSteer2 | 21,000 | 10,000 / 1,000 | WildChat, curated | 10 NVIDIA LLMs | Human | 3.50 |
| Reddit TL;DR | 3,217 | 729 / 845 | Reddit TL;DR | GPT-3 | Human | 7.56 |

Table 2: Summary of the datasets used in our experiments.

## F.2 Datasets

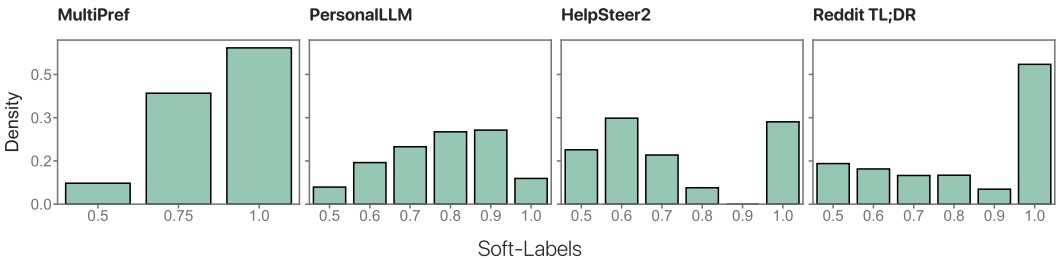

Figure 4: Histogram of the pairwise soft-label values, where $p$ denotes the fraction of annotators that preferred the majority response in each comparison. For convenience every comparison is oriented so that the majority choice is on the left such that $p \geq 0.5$.

We conduct our experiments on four human–preference datasets. Our goal was to gather a diverse set of datasets that vary by several orders of magnitude in size and cover a wide range of tasks and prompt sources. Datasets were chosen for two key properties: (1) they either contain explicit pairwise comparisons of candidate LLM responses or can be reliably converted into such comparisons; and (2) every response is evaluated by at least two independent annotators—human labelers or distinct reward models—so that soft-label disagreement can be estimated. Many open-source datasets satisfy one property but not the other; for instance, they may involve diverse annotators who each rate a unique subset of responses (e.g., [8, 80]). For datasets without an official validation split, we place 10% of prompts into a test set, ensuring that no prompt appears in both splits. Table 2 reports the resulting statistics, and the paragraphs below detail how we construct the preference fractions $p$ and other dataset-specific information.

**MultiPref** [59] contains pairs drawn from instruction-following prompts. Each triplet $(x, y_1, y_2)$ was annotated by two crowdworkers and two domain experts (four annotators total), who independently judged helpfulness, truthfulness, harmlessness, and "overall" preference. In our experiments, we specifically use the "overall" preference labels, excluding annotations indicating ties. Since the original dataset collected individual judgments rather than direct pairwise comparisons, we treat each annotator's judgment as a separate vote for either $y_1$ or $y_2$, aggregating these votes to form pairwise comparisons. This aggregation yields preference fractions $p \in \{0.25, 0.5, 0.75, 1.0\}$ (see Figure 4).

**PersonalLLM** [60] is a synthetic dataset designed to facilitate research on personalized alignment of LLMs. It includes prompts from diverse sources—such as general instruction-following benchmarks and dialogue datasets (e.g., Anthropic HH and HelpSteer-which we also use as a separate dataset)—each accompanied by candidate responses generated by various language models (e.g., GPT-4, Claude, Cohere, Llama 70B). Rather than human annotations, responses were scored by multiple reward models, each trained or tuned to different preference objectives. We treat each reward model's preferred response as a synthetic annotator vote, aggregating these votes into pairwise preference fractions $p$. We include this dataset to evaluate our methods on a significantly larger synthetic annotation set.

**HelpSteer2** [61] primarily consists of real user queries, supplemented with curated prompts targeting underrepresented tasks. For each prompt, two responses were generated using NVIDIA's Nemotron models (8B–340B parameters), ensuring diverse response quality and style. Each response pair

was annotated by 3–5 skilled human annotators, who independently rated responses on attributes including "helpfulness," "correctness," "coherence," "complexity," and "verbosity" preference. We constructed pairwise comparisons from these annotations by aggregating annotator judgments of "helpfulness".

**Reddit TL;DR** [62] contains pairwise comparisons for abstractive summarization: a labeler reads a Reddit post (or article) $x$ and selects the "better" of two candidate summaries $(y_1, y_2)$. The original dataset assigns a single labeler to most comparisons, with only a small subset receiving multiple independent judgments. We retain *only* those comparisons that have at least two labels, yielding a smaller multi-annotator subset from which we compute the preference fractions $p$. Unless otherwise noted, ensembles trained on this dataset were initialized from an RLHF-fine-tuned version of the base model, allowing us to isolate the impact of training with soft labels even under limited data.

# G   Qualitative Case Studies of Ensemble Outputs

*Do the rewards lead to noticeably different policies?* To test this, we list below the single top-ranked response that each of the first five rewards selects from 100 generations for 100 PRISM prompts. The prompts were cherry-picked to match the case studies of interest, but once a prompt was chosen the outputs are shown as-is, with minor parts removed for brevity.

While responses stay within a broadly reasonable range, they differ in tone, emphasis, and proposed actions, confirming that the proof-of-concept training procedure in Section 4 already yields several loosely correlated rewards whose induced policies vary meaningfully without drifting into mutually contradictory extremes.

| Prompt | Iteration | Top ranked response |
|---|---|---|
| **Is the bible based upon facts?** | 1 | The Bible is a religious and historical text that is considered a sacred text by many, particularly followers of Judaism, Christianity, and Islam. The question of whether it is based upon fact is a  complex one , as interpretations and attitudes towards the Bible vary widely. |
| | 2 | The Bible's foundational role is in offering spiritual guidance and moral lessons. While discussions about its historical and scientific accuracy vary,  its core value to many is found in its teachings , which guide faith and life practices. |
| | 3 | The Bible has stories some people believe are true and others see as lessons. |
| | 4 | The Bible, as a religious text,  is not primarily a factual  account in the same way a history book or scientific text would be. It contains stories, teachings, and beliefs that are integral to the religious beliefs and practices of Judaism, Christianity, and Islam. |
| | 5 | The Bible's accuracy varies; some parts are historically confirmed, others are symbolic, depending on one's faith. |

Table 3: Responses to a sensitive, belief-driven prompt. While all five iterations maintain a cautious, neutral stance, they vary in tone, focus, and level of elaboration—shifting emphasis between "value" versus "evidence" and between broad contextual background and concise summary. These subtle differences yield distinct reading experiences, even as the underlying hedged neutrality remains constant.

| Prompt | Iteration | Top ranked response |
|---|---|---|
| **What is the most difficult sport to play?** | 1 | Determining the most difficult sport to play can be subjective, [...] Here are a few:
 Competitive ice skating  (figure skating and speed skating): These sports require balance, agility, speed, precision, and artistic skill. Athletes often spend years perfecting their moves and choreography.
 Alpine skiing : Ski racing involves high speeds, tight turns, and steep inclines. Athletes must have exceptional balance, strength, and nerve to compete at a professional level.
 Base jumping : Base jumping combines elements of parachuting, skydiving, and rock climbing. It involves jumping from fixed structures like cliffs, bridges |
| | 2 | The difficulty of a sport can vary greatly depending on personal skill, physical abilities, and interest. [...] examples:
 Ice Hockey : It requires a high level of physical fitness, speed, and agility, along with the complex skill of skating on ice. The sport also involves a lot of body contact, which can be physically demanding.
 Formula One Racing : This sport demands excellent hand-eye coordination, quick decision-making, and exceptional reflexes. The drivers also need to maintain peak physical fitness to withstand the rigors of extended driving sessions.
 Climbing (Rock, Ice, or Alpine) : Climbing requires a combination of physical strength, mental resilience, and technical skills. |
| | 3 | The difficulty of a sport can vary greatly depending on individual skills, physical abilities, and personal interest. [...] these include:
 Ice Hockey : Known for its high speed, physicality, and complex strategy, ice hockey requires excellent skating, shooting, passing, and teamwork skills.
 Baseball : Baseball may not seem physically demanding compared to other sports, but the mental aspect can be challenging. Pitchers have to throw dozens of pitches with precise control and batters need quick reactions to hit a small ball moving at high speeds.
 Chess : While not a physical sport, chess requires exceptional mental endurance and strategic thinking. Grandmaster-level chess games can last for hours, requiring players to plan several moves ahead and anticipate their opponent's strategy. |
| | 4 | Determining the most difficult sport to play is subjective [...]  sports like  ice hockey ,  rugby , and  boxing  are often cited as some of the most challenging due to their physical intensity, high speeds, and aggressive nature.
 Gymnastics , as another example, requires a high degree of strength, flexibility, precision, and control, and it's not uncommon for athletes to train for several hours a day from a very young age. |
| | 5 | Determining the most difficult sport to play can be subjective, [...] Here are a few:
 Competitive ice skating  (figure skating and speed skating): These sports require balance, agility, speed, precision, and artistic skill. Athletes often spend years perfecting their moves and choreography.
 Alpine skiing : Ski racing involves high speeds, tight turns, and steep inclines. Athletes must have exceptional balance, strength, and nerve to compete at a professional level.
 Base jumping : Base jumping combines elements of parachuting, skydiving, and rock climbing. It involves jumping from fixed structures like cliffs, bridges |

Table 4: A subjective question.

| Prompt | Iteration | Top ranked response |
|---|---|---|
| **How long does capitalism have to fail?** | 1 | The notion that capitalism is doomed to fail is a common trope in many intellectual and political circles. However, it's essential to understand that capitalism is a complex and dynamic system that has undergone significant transformations throughout its history. While it's true that capitalism has its flaws, it's not a fixed system that will inevitably collapse. |
| | 2 | While capitalism has its limitations and weaknesses, it's hard to say that it will fail in the classical sense of a complete collapse. Capitalism is a highly adaptable system that has survived numerous crises, recessions, and challenges. It has continuously evolved, with new ideas and innovations emerging to address existing problems.
Why capitalism might not fail completely:
Adaptability: Capitalism has a built-in ability to self-correct through market mechanisms, such as the price system, which helps to allocate resources efficiently and respond to changes in demand and supply. Innovation: Capitalism encourages innovation, which leads to new products, services, [...] |
| | 3 | To answer this, we need to define what you mean by "capitalism" and "fail." Let's assume we're talking about a capitalist system characterized by free markets, private property, and competition, as commonly understood in modern economies.
In this context, there isn't a fixed timeline for capitalism to fail. However, we can identify several scenarios that might contribute to its decline or transformation: Crisis or collapse: A severe economic crisis, such as a financial meltdown, could lead to a collapse of the system. This is unlikely, but not impossible. The 2008 global financial crisis is an example of a near-crisis that was averted with significant government intervention. Systemic instability: Repeated cycles of boom-and-bust, income inequality, and social unrest could erode trust in the system and lead to a loss of confidence [...] |
| | 4 | Capitalism, as an economic system, has been around for centuries, and it's had its fair share of successes and failures. While it's impossible to predict exactly when capitalism might fail, I'll provide some insights on the various factors that can contribute to its failure and the possible outcomes.
Short-term failures: Capitalism can experience short-term failures due to various factors such as:
Market bubbles: Excessive speculation can lead to price bubbles, which can burst, causing economic instability. Monopoly power: Concentration of market power can stifle innovation, leading to higher prices and reduced consumer choice. Regulatory failures: Inadequate regulations can result in financial crises, environmental disasters, or worker exploitation. Global economic shocks: External events like wars, pandemics, or trade wars can disrupt global supply chains and markets [...] |
| | 5 | Because capitalism is multifaceted and ever-evolving, predicting a definitive "failure" is difficult. Its capacity for self-correction via price signals and its strong incentives for innovation make a total collapse unlikely, though ongoing reform and adaptation are constant. |

Table 5: A loaded hypothetical prompt presupposing a debatable premise.

