# OpenReview forum: "Pairwise Calibrated Rewards for Pluralistic Alignment"
_NeurIPS.cc/2025/Conference — NeurIPS 2025 poster_

### Official Review · Reviewer_sbAB · 2025-06-30

**Clarity:** 4
**Significance:** 3
**Originality:** 3
**Rating:** 5
**Confidence:** 4

**Summary:**

This paper studies pluralistic alignment, the problem of aligning models when desired behavior is multi-modal or subjective. The paper proposes a reward ensemble model to capture this multi-modality, which is trained to minimize the calibration error wrt different annotator preferences. The goal is to find a small set of ensembles that capture this multi-modality and practical to train. Before giving a training algorithm, the authors prove that even under a simplified setting, this problem is NP-hard; but a well-calibrated ensemble approximation is possible. Outlier reward functions are possible but their total weights decay as reciprocal of their relative disagreement ($1/(\beta-1)$). Rather than fitting the ensemble model directly, which is computationally intensive, the authors propose using forward stagewise additive modeling approach that decomposes the problem into a sequence of easier-to-solve subproblems. First, a base reward function (modeled as a reward head attached to a frozen SFT-ed model) is fitted to the calibration probabilities and residuals are estimated. A newly initialized reward function is fitted to the residuals and this process is repeated for $k$ iterations. Experimental results suggest that compared to theoretically best single reward function, ensemble approach achieves lower MSE error on unseen prompts and different reward models capture distinct preferences.

**Questions:**

I would increase my score when the followings are addressed.

1. Please run an LLM post-training experiment with the proposed reward ensemble.

2. Please add reward ensemble baselines. It could just be optimizing Eq (2) jointly over all reward functions and weights.

3. Is using $\sigma$ for minimizing MSE over residuals appropriate?

4. What is the impact of increasing number of annotators?

5. What is the impact of temperature on correlation and MSE loss?

6. Is updating all $\alpha$s after every step necessary to get good performance?

7. Please add MSE loss on training prompts.

**Ethical Concerns:**

["NO or VERY MINOR ethics concerns only"]

**Final Justification:**

The authors adressed most of my concerns and added additional empirical results for improving the analysis.

**Limitations:**

Yes.

**Quality:**

3

**Strengths And Weaknesses:**

# Strengths
1. **Theoretically justified approach to multi-modal preference modeling.**
The paper clearly explains the pluralistic alignment problem with a detailed motivation, contrasting with previous approaches. It introduces pairwise calibration for reward model ensembles and shows that finding $\epsilon$-calibrated ensembles is NP-hard. Given a reward model ensemble is pairwise calibrated, it proves its robustness -- outliers have bounded reward weights.

2. **A practical algorithm for optimizing reward ensemble objective.**
Full optimization problem involves tuning $k$ reward models as well as their weights. This objective can become computationally intensive. The paper proposes using forward stagewise additive modeling (FSAM), an approach for learning ensembles, for learning reward ensembles and their weights. While fitting a fresh reward model to residuals is similar to the original FSAM, the paper retunes weights at each step.

3. **Experimental results show superiority to single model reward functions and diversity.**
Experimental results using four different datasets show that ensemble approach achieves lower MSE on holdout set and produces diverse reward models when measured using Kendal-$\tau$ on rankings.

# Weaknesses
1. **No LLM post-training with the proposed ensemble approach.** While the paper clearly explains why the ensemble approach is needed for better LLM alignment and briefly touches on how reward function ensembles can be used during inference, it doesn't provide any quantitative metrics on whether it actually is useful in practice. I think at least the performance of BoN with the trained reward ensemble should be reported and compared to baselines.

2. **Lack of ensemble baselines.** While theoretical single reward limit is useful to understand the relative improvement of the proposed model, no other ensemble baselines is used in the comparison. For example, can you add multiple-reward heads and optimize objective in Eq. (2) jointly as an oracle?

3. **Implication of using $\sigma$.** While the first reward model is learned using probabilities (preference fractions) as targets, other reward models are learned using residuals -- difference of two probabilities -- as targets. These two are of different units. While a normalized function (i.e. $\sigma$) is used in both cases, it is only appropriate for the first reward model training.

4. **The impact of the number of annotators.** How much does the number of annotators per prompt impact the overall results? What happens if you have only 2 annotators? Please provide additional results by resampling (without replacement) responses for each datasets to simulate increasing number of annotators (from 2 to more).

5. **The impact of Temperature on Kendall-$\tau$ and MSE.** How much does the temperature impact the overall results? Please provide additional empirical results by varying the temperature and measuring Kendall-$\tau$ and MSE loss.

6. **The impact of updating $\alpha$ vs keeping them fixed.** What happens if you keep $\alpha$s fixed after each step rather than tuning them again?

7. **MSE on training prompts.** Please add MSE on training prompts in Figure 2.

---

> ### Author Rebuttal · Authors · 2025-07-30
>
> >  No LLM post-training with the proposed ensemble approach. While the paper clearly explains why the ensemble approach is needed for better LLM alignment and briefly touches on how reward function ensembles can be used during inference, it doesn't provide any quantitative metrics on whether it actually is useful in practice. I think at least the performance of BoN with the trained reward ensemble should be reported and compared to baselines.
>
> > Please run an LLM post-training experiment with the proposed reward ensemble.
>
> Could you please outline a concrete post-training experiment we could run? Due to the nature of our approach, quantitative evaluation of our reward ensemble is challenging in practice because the ensemble is designed to preserve several plausible answers rather than collapse them into one. Conventional single-output evaluations common in the literature, such as win-rate or average reward, require a canonical answer or a monolithic notion of “better” and therefore obscure the benefit of capturing multiple annotator perspectives.
>
> Prior work that evaluates multiple policies typically assumes a known ground-truth distribution over answers or explicit group labels for annotators. Our framework works without group identities, finding a small ensemble whose voting pattern matches the observed pairwise fractions, so evaluation methods that reduce to binary wins or accuracies do not transfer directly.
>
> In summary, if you can point us to an evaluation method that respects multiple plausible outputs, we will gladly run it and include the results.
>
> >  Lack of ensemble baselines. While theoretical single reward limit is useful to understand the relative improvement of the proposed model, no other ensemble baselines is used in the comparison. For example, can you add multiple-reward heads and optimize objective in Eq. (2) jointly as an oracle?
>
> Optimizing the joint objective in Eq. (2) over all reward parameters and mixture weights is highly nonconvex, and we are not aware of a standard algorithm that would serve as a clear canonical baseline. Our empirical evaluation was designed to show feasibility: a simple greedy heuristic can recover a well-calibrated ensemble on real preference data, demonstrating that small reward ensembles are practical rather than only a theoretical construct. Developing and benchmarking more advanced optimization techniques for this loss is an exciting direction for future work. If you have a specific joint-training approach you consider appropriate, we would be happy to try it and report the results.
>
> >  Implication of using $\sigma$ and $\alpha$. While the first reward model is learned using probabilities (preference fractions) as targets, other reward models are learned using residuals -- difference of two probabilities -- as targets. These two are of different units. While a normalized function (i.e. $\sigma$) is used in both cases, it is only appropriate for the first reward model training.
>
> >  The impact of updating $\alpha$ vs keeping them fixed. What happens if you keep $\alpha$s fixed after each step rather than tuning them again?
>
>
> Thank you for raising these two points. First, we experimented with three weighting strategies and observed only modest differences in calibration. The first strategy freezes the existing mixture at round $j$, learns a new weight $\alpha_j$, and rescales earlier weights by $(1-\alpha_j)$. The second strategy sets $\alpha_j$ to $1/j$ to create an unweighted running average. The third strategy, our default, retrains the entire weight vector after adding each new head. For example, with $k=8$ on *MultiPref*, the test MSEs were 0.071 for the fully reweighted mixture, 0.073 for the fixed‑previous scheme, and 0.081 for the unweighted average. The refitted mixtures typically converge to weights close to $1/j$, suggesting that equal weighting is already near optimal and further optimization only makes minor adjustments.
>
> Second, while the first reward head is trained on preference fractions and the later heads on residuals, the subsequent weight update absorbs the scale difference introduced by the sigmoid. In practice the combined model remains well calibrated because the learned mixture rebalances the contributions of earlier and later heads.
>
> >  The impact of the number of annotators. How much does the number of annotators per prompt impact the overall results? What happens if you have only 2 annotators? Please provide additional results by resampling (without replacement) responses for each datasets to simulate increasing number of annotators (from 2 to more).
>
> Thank you for raising this point. We agree that the number of annotators per prompt could influence calibration and are running additional tests. As a first step, we resampled the PersonalLLM dataset with five annotators instead of the original ten. Preliminary results (see below) compare $n = 5$ against $n = 10$. We will extend this analysis and report the full results in the next revision.
>
> |  round_inx  |   mse_n10 |   mse_n5 |
> |---:|----------:|---------:|
> |  0 |     0.097 |    0.104 |
> |  1 |     0.078 |    0.060 |
> |  2 |     0.049 |    0.049 |
> |  3 |     0.038 |    0.049 |
> |  4 |     0.034 |    0.047 |
> |  5 |     0.032 |    0.046 |
> |  6 |     0.031 |    0.045 |
> |  7 |     0.031 |    0.044 |
>
>
> > The impact of Temperature on Kendall-$\tau$ and MSE. How much does the temperature impact the overall results? Please provide additional empirical results by varying the temperature and measuring Kendall-$\tau$ and MSE loss.
>
> To test whether the reward models in the ensemble capture distinct preferences, we evaluate how they rank multiple candidate responses to the same prompts. We need a diverse pool of answers for this comparison, so we control diversity with the sampling temperature and experimented with values between 0.7 and 1.2.
>
> When the temperature is low, the model produces almost identical answers (semantically and stylistically). Near-duplication collapses the effective sample size and can amplify noise as the scores assigned by a reward function to near-duplicates differ only by small random fluctuations. A higher temperature injects enough diversity that score gaps reflect genuine judgment differences rather than tie-breaking randomness.
>
> Below we attach additional results showing the effect of temperature on the Kendall-τ metric for the *MultiPref* dataset:
>
> * *T=1.2*:
>
> |         |   Round 0 |   Round 1 |   Round 2 |   Round 3 |   Round 4 |   Round 5 |   Round 6 |   Round 7 |
> |:--------|----------:|----------:|----------:|----------:|----------:|----------:|----------:|----------:|
> | Round 0 |     1.000 |    -0.066 |     0.200 |     0.162 |     0.442 |     0.237 |     0.453 |     0.469 |
> | Round 1 |    -0.066 |     1.000 |    -0.275 |     0.254 |    -0.328 |    -0.144 |    -0.311 |    -0.286 |
> | Round 2 |     0.200 |    -0.275 |     1.000 |    -0.047 |     0.247 |     0.108 |     0.356 |     0.279 |
> | Round 3 |     0.162 |     0.254 |    -0.047 |     1.000 |    -0.038 |     0.002 |    -0.039 |     0.023 |
> | Round 4 |     0.442 |    -0.328 |     0.247 |    -0.038 |     1.000 |     0.260 |     0.600 |     0.692 |
> | Round 5 |     0.237 |    -0.144 |     0.108 |     0.002 |     0.260 |     1.000 |     0.231 |     0.213 |
> | Round 6 |     0.453 |    -0.311 |     0.356 |    -0.039 |     0.600 |     0.231 |     1.000 |     0.646 |
> | Round 7 |     0.469 |    -0.286 |     0.279 |     0.023 |     0.692 |     0.213 |     0.646 |     1.000 |
>
> * *T=1.0*:
>
>
> |         |   Round 0 |   Round 1 |   Round 2 |   Round 3 |   Round 4 |   Round 5 |   Round 6 |   Round 7 |
> |:--------|----------:|----------:|----------:|----------:|----------:|----------:|----------:|----------:|
> | Round 0 |     1.000 |    -0.133 |     0.205 |     0.221 |     0.272 |     0.159 |     0.339 |     0.220 |
> | Round 1 |    -0.133 |     1.000 |    -0.268 |     0.162 |    -0.235 |    -0.066 |    -0.127 |    -0.052 |
> | Round 2 |     0.205 |    -0.268 |     1.000 |    -0.122 |     0.096 |     0.078 |     0.297 |     0.060 |
> | Round 3 |     0.221 |     0.162 |    -0.122 |     1.000 |     0.097 |    -0.032 |     0.027 |     0.193 |
> | Round 4 |     0.272 |    -0.235 |     0.096 |     0.097 |     1.000 |     0.091 |     0.232 |     0.393 |
> | Round 5 |     0.159 |    -0.066 |     0.078 |    -0.032 |     0.091 |     1.000 |     0.105 |     0.032 |
> | Round 6 |     0.339 |    -0.127 |     0.297 |     0.027 |     0.232 |     0.105 |     1.000 |     0.268 |
> | Round 7 |     0.220 |    -0.052 |     0.060 |     0.193 |     0.393 |     0.032 |     0.268 |     1.000 |
>
> * *T=0.8*:
>
> |         |   Round 0 |   Round 1 |   Round 2 |   Round 3 |   Round 4 |   Round 5 |   Round 6 |   Round 7 |
> |:--------|----------:|----------:|----------:|----------:|----------:|----------:|----------:|----------:|
> | Round 0 |     1.000 |    -0.524 |     0.297 |    -0.260 |     0.561 |     0.487 |     0.523 |     0.461 |
> | Round 1 |    -0.524 |     1.000 |    -0.371 |     0.420 |    -0.616 |    -0.484 |    -0.445 |    -0.421 |
> | Round 2 |     0.297 |    -0.371 |     1.000 |    -0.288 |     0.298 |     0.297 |     0.339 |     0.277 |
> | Round 3 |    -0.260 |     0.420 |    -0.288 |     1.000 |    -0.326 |    -0.327 |    -0.308 |    -0.238 |
> | Round 4 |     0.561 |    -0.616 |     0.298 |    -0.326 |     1.000 |     0.477 |     0.438 |     0.515 |
> | Round 5 |     0.487 |    -0.484 |     0.297 |    -0.327 |     0.477 |     1.000 |     0.469 |     0.400 |
> | Round 6 |     0.523 |    -0.445 |     0.339 |    -0.308 |     0.438 |     0.469 |     1.000 |     0.389 |
> | Round 7 |     0.461 |    -0.421 |     0.277 |    -0.238 |     0.515 |     0.400 |     0.389 |     1.000 |
>
> > MSE on training prompts. Please add MSE on training prompts in Figure 2.
>
> Thank you for pointing this out, we will add the MSE on training prompts to Figure 2 in the revised manuscript.

---

> > ### Author Response · Authors · 2025-08-07
> >
> > Our rebuttal contains a number of new empirical results and we believe it addresses most (if not all) of your concerns and questions. We would greatly appreciate the chance to discuss the rebuttal with you.

---

> > ### Comment · Reviewer_sbAB · 2025-08-09
> >
> > My point about empirical results was mainly about understanding the empirical impact of the proposed reward ensemble during inference, such as running BoN algorithm against any practical reward function that your ensemble suggests. While I still would like to see some empirical results on usefulness of your method, my remaining concerns are adressed. I updated my score accordingly.

---

### Official Review · Reviewer_cPuB · 2025-07-02

**Clarity:** 3
**Significance:** 4
**Originality:** 4
**Rating:** 5
**Confidence:** 3

**Summary:**

RLHF mostly represents the aggregate view of the majority, dismissing the inherent disagreement and diverging opinions across the population. This limits the utility of current models. This paper proposes a methodology to move to a distribution over reward-functions to reflect the diversity of judgments. For each reward, a LLM is fine-tuned which results in a distribution over policies. An ensemble of reward functions is formed to signify the diverging human preferences.  The methodology aims to be calibrated pairwise; for every pairwise preference, the fraction of reward functions in the ensemble “voting” for a preference matches the fraction of annotators with that preference.

The authors first formally define pairwise calibration and provide theoretical guarantees for getting pairwise calibrated ensembles with a small support, for how removing outlier rewards has minimal impact on the calibration, and that the ensembles can generalize to unseen populations.

The method was tested empirically on four pairwise comparison based datasets and the calibration was evaluated through Mean-Squared Error calibration, showing reduced MSE as the ensemble increases to around 2-4.

**Questions:**

- Why did you opt for MSE loss specifically? And e.g., not a soft cross-entropy loss or JSD loss?
- Is this setup also extendable to tasks that are not set up in a pairwise preference way?
- Results indicate that from 2 reward models onwards, the calibration improves on held-out prompts, do you think this will extend to other tasks as well? Do you have an intuition of how this will change based on the subjectivity of the task?
- Is there a specific reason you did not include calibration metrics such as Brier Score or ECE?
- Do the pairwise Kendall $\tau$ correlation scores in Figure 3 also reflect and explain the ensemble size, e.g., that only 2 reward models are already sufficient?

**Ethical Concerns:**

["NO or VERY MINOR ethics concerns only"]

**Final Justification:**

Given the novelty of the task the paper is addressing, pairwise calibration for pluralistic alignment, I have decided to raise my score. Many of my concerns were adequately addressed:
- The authors indicate that they will improve the related work section. This especially made me raise my score as for me it was unclear how to position this paper in this subarea of pluralistic alignment and why there were no baselines. To me it is now clear that this is one of the first work (that I am aware of) that addresses pairwise calibration in reward models. I did try to think of any existing baselines that could fit for this setup but could not think of any, it is possible I might have missed something.
- The point regarding unseen datasets was clarified.
- The authors indicated that they will discuss more about undesirable viewpoints and annotator noise.

I do think the authors could still add a small analysis regarding the performance of the model as disagreement level changes.

I also took into account the responses to the author reviewers when raising my score.

**Limitations:**

yes

**Quality:**

3

**Strengths And Weaknesses:**

**Strengths.**
- The work tackles an important problem and does that through a well motivated approach. I really like how it is supported by theoretical guarantees.
- The paper was easy to read and looked neat.
- In a way, it is good that this method does not rely on explicitly assigning reward models to specific demographics to represent the population. This makes the method more adaptable as making the decision which to include and which not makes it difficult with annotator availability, intersectionality, etc.

**Weaknesses.**
- The positioning of the paper in comparison to other related work is difficult for the reader. The Related Work is in the Appendix and a highly condensed version in the main body did not give a clear idea. This is also reflected in the Empirical Results section where there is no comparison to other baselines. As such, it is hard to get an idea of how much improvement this pairwise calibration brings. If there are no directly related baselines, maybe it is a good idea to make this more prominent in the Related Work section.
- There are no experiments where the method is applied to unseen datasets. This would really strengthen the paper as it could help in understanding whether the method does the task well or it might actually just be overfitting to the annotators of a dataset (some might suffer from a lack of diversity of annotators).
- I find that the discussion surrounding undesirable viewpoints or outliers could be a bit more prominent. When is a viewpoint or preference undesirable? Is this e.g., hateful responses? This should be clarified a bit. Currently, noisy annotations (an annotator might be making mistakes) are not discussed whether their method would be robust to that or not or whether these fall under outliers.
- The paper does not test datasets that have a lower agreement rate around 50%, or does an analysis of how performance is in terms of different levels of agreement across samples.

---

> ### Author Rebuttal · Authors · 2025-07-30
>
> > * The positioning of the paper in comparison to other related work is difficult for the reader. The Related Work is in the Appendix and a highly condensed version in the main body did not give a clear idea. This is also reflected in the Empirical Results section where there is no comparison to other baselines. As such, it is hard to get an idea of how much improvement this pairwise calibration brings. If there are no directly related baselines, maybe it is a good idea to make this more prominent in the Related Work section.
>
> We agree that clearer positioning is warranted. The main submission condensed the related work due to the page limit, but we will move key context from the appendix into the main paper to address your comment. Below we briefly elaborate on the lack of baselines.
>
> Existing multi‑reward techniques hinge on predefined groups, inferred clusters, or annotator identifiers and therefore assume the very structure that our method seeks to avoid. Because pairwise calibration learns directly from unlabeled disagreement, there is no off‑the‑shelf baseline that operates under the same information constraints.
>
> Constructing a head‑to‑head comparison would require first partitioning annotators or fabricating demographic tags, which would inject assumptions our approach is designed to remove.
>
> This absence of compatible baselines is inherent to the contribution: we introduce a framework that treats disagreement as signal without relying on latent or explicit clusters, filling a gap left open by prior group‑conditioned methods.
>
>
> > * There are no experiments where the method is applied to unseen datasets. This would really strengthen the paper as it could help in understanding whether the method does the task well or it might actually just be overfitting to the annotators of a dataset (some might suffer from a lack of diversity of annotators).
>
> Generalization can be understood along several axes: new responses, unseen prompts, additional annotators, and different task types. Our theoretical analysis in Section 3.3, roughly speaking, shows that the ensemble inherits the same sample-complexity guarantees as its constituent reward models, supporting its ability to extend beyond the training comparisons.
>
> The datasets we analyze already probe several of these axes. For example, *MultiPref* and *HelpSteer2* draw feedback from about $300$ and $1,000$ unique annotators, respectively, so each evaluation fold mixes many human viewpoints rather than revisiting the same few raters. Moreover, *MultiPref* and *PersonalLLM* combine data from diverse task domains such as open-ended question answering, creative writing, code generation, and safety probes, so our held-out evaluations test the ensemble on a broad range of goals instead of a single task.
>
>
> > * I find that the discussion surrounding undesirable viewpoints or outliers could be a bit more prominent. When is a viewpoint or preference undesirable? Is this e.g., hateful responses? This should be clarified a bit.
>
> We agree that a clearer discussion of “outliers” will strengthen the paper. Conceptually, learning multiple viewpoints risks allocating weight to preferences that most users would deem undesirable, such as hateful responses. In our terminology, an outlier is a reward function whose judgments diverge sharply from the rest of the population. We measure this via the disagreement score (line 156): a reward is considered extreme when its score is large relative to the minimum attainable value. Admittedly, according to this definition, not every outlier viewpoint is repugnant, but every repugnant viewpoint is an outlier, and the latter implication is the one we conceptually rely on to ensure the ensemble is safe.
>
>
> > Currently, noisy annotations (an annotator might be making mistakes) are not discussed whether their method would be robust to that or not or whether these fall under outliers.
>
> Although we do not model label noise explicitly, the ensemble method is especially valuable in settings where a few annotation errors could flip the majority label and mislead a single-reward approach. Because the ensemble tracks the full preference fraction rather than a hard majority, a stray vote merely adjusts the target probability slightly, so no single mistake dominates the learning signal.
>
> > * The paper does not test datasets that have a lower agreement rate around 50%, or does an analysis of how performance is in terms of different levels of agreement across samples.
>
> We quantify disagreement for every comparison by $p$, the fraction of annotators who choose the majority answer, where $p = 0.5$ is the highest possible disagreement. The datasets we study span a wide range of $p$ values, including many low-agreement cases.
>
> Furthermore, the small numbers of annotators per pair can introduce an upward bias in the observed agreement value, meaning our datasets may have more disagreement than the observed $p$ reveals. To illustrate, consider a scenario of maximal disagreement where the true population preference is split 50/50 on every pair. If we sample only four annotators for each, the observed majority fraction for each edge will be 1.0, 0.75, or 0.5 with respective probabilities of 1/8, 1/2, and 3/8. This results in an expected value of 0.69.
>
> **Questions**
> >  * Why did you opt for MSE loss specifically? And e.g., not a soft cross-entropy loss or JSD loss?
>
> We chose MSE for analytical convenience and cleaner theory, and we expect other proper divergences that penalize discrepancies between $\hat p$ and $p^\star$ would behave similarly in principle. For the algorithm, residuals under MSE can fall outside the $[0,1]$ range, giving each subsequent reward a stronger signal to close the remaining calibration gap.
>
> > * Is this setup also extendable to tasks that are not set up in a pairwise preference way?
>
> Thank you for raising this. As briefly touched upon in Section 6, our analysis focuses on pairwise preferences and does not guarantee an accurate representation of higher-order judgments over larger candidate sets. Ideally, for any group of responses, the share of learned reward functions ranking a particular answer first would reflect the true pattern of preferences in the population. Recovering that structure from pairwise data alone is information-theoretically impossible because such comparisons lack enough detail to pin down full rankings. Capturing these richer relationships would require more expressive feedback, such as complete rankings or best-of-$\ell$ choices drawn from larger pools of answers. Exploring how to extend the framework to those settings is an exciting direction for future work.
>
> >  * Results indicate that from 2 reward models onwards, the calibration improves on held-out prompts, do you think this will extend to other tasks as well? Do you have an intuition of how this will change based on the subjectivity of the task?
> > * Do the pairwise Kendall correlation scores in Figure 3 also reflect and explain the ensemble size, e.g., that only 2 reward models are already sufficient?
>
> Intuitively, we expect the trend to generalize. Tasks with higher subjectivity, such as creative writing style or humor, should keep gaining from additional reward models, because each model captures different evaluator biases. For more objective tasks, most evaluators share the same preference ordering, so the first model already captures the key signal, extra models mostly fit residual noise, and calibration gains taper off after one or two models.
>
> >  * Is there a specific reason you did not include calibration metrics such as Brier Score or ECE?
>
> Thank you for raising this. The Brier score is effectively the mean squared error between the predicted probability and the observed outcome, which we already report. We will include ECE in the next revision to give an additional view of calibration.

---

> > ### Comment · Reviewer_cPuB · 2025-08-05
> >
> > Thank you for addressing my concerns and answering my questions.
> >
> > Most of the weaknesses I raised have been resolved to a large extent. I understand that there is no baseline due to the novelty of what this paper is trying to address. I do think this can be made clearer in the Related Work which the authors have indicated they will make more elaborate in the next version of the paper.
> >
> > It would be great if you could add the points you mentioned regarding the outliers and annotation noise to the paper!
> >
> > Regarding the $p$-value, thank you for clarifying! I still think a small analysis of how the performance looks like across different levels of agreement would still be interesting, if possible.
> >
> > I have decided to raise my score.

---

> > > ### Author Response · Authors · 2025-08-05
> > >
> > > Thank you for your response. We'd be happy to address these points in the revised version of the paper.

---

### Official Review · Reviewer_fob7 · 2025-07-06

**Clarity:** 3
**Significance:** 3
**Originality:** 3
**Rating:** 4
**Confidence:** 2

**Summary:**

The paper introduces a novel approach to alignment by replacing a single reward model with a distribution over reward functions, enabling more flexible and pluralistic AI behavior. It avoids rigid annotator groupings by learning compact, diverse reward ensembles directly from preference data, achieving better calibration and preserving minority viewpoints. However, the method raises concerns about scalability, the challenge of selecting the right number of rewards, and whether such complexity is always necessary compared to simpler, single-reward models.

**Questions:**

Please see my comments above.

**Ethical Concerns:**

["NO or VERY MINOR ethics concerns only"]

**Final Justification:**

I have no further technical questions to the paper. I am inclined to rate it as a weak accept as reflected by my original score.

**Limitations:**

Yes.

**Paper Formatting Concerns:**

No.

**Quality:**

3

**Strengths And Weaknesses:**

# Strengths
- The paper replaces the single-reward model with a distribution over reward functions, each reflecting a different interpretation of human judgment. By fine-tuning a separate LLM for each reward, it produces a distribution over policies. This enables flexible inference—offering diverse viewpoints, user-specific responses, or inclusive summaries.
- The paper learns a compact ensemble of reward functions from aggregated preferences, without using annotator IDs or predefined groups. It introduces pairwise calibration to ensure the ensemble reflects the distribution of annotator opinions. This avoids preference collapse and preserves pluralism without rigid clustering.
- The paper also demonstrates that compact, diverse reward ensembles can be efficiently learned using a forward stage-wise additive modeling process. Experiments show these ensembles achieve lower calibration MSE than single-reward models, validating their ability to represent diverse preferences accurately.

# Weakness
While the paper presents a compelling approach by replacing the single-reward model with a distribution over diverse reward functions and training separate LLMs for each, there are still several concerns regarding the method.
- Determining the number of reward functions: how should the appropriate number be chosen, and what are the consequences of selecting too few or too many?
- Computational cost: training and maintaining multiple policies significantly increases resource requirements. Are there strategies to mitigate this overhead?
- Practical necessity: in many cases, a single reward function may suffice. How can we assess when it's truly beneficial to adopt multiple reward models?

---

> ### Author Rebuttal · Authors · 2025-07-30
>
> > * Determining the number of reward functions: how should the appropriate number be chosen, and what are the consequences of selecting too few or too many?
> > * Practical necessity: in many cases, a single reward function may suffice. How can we assess when it's truly beneficial to adopt multiple reward models?
>
> We view the ensemble size as a balance between capturing diversity and practical deployment concerns. If the ensemble is too small, it can under‑represent minority viewpoints. If it grows beyond what the task needs, it introduces redundant parameters, increases training and inference cost, and may steer the system toward unwanted personalization.
>
> A single reward can indeed suffice in some cases. Yet, when users tend to prefer different outputs or when output diversity in downstream applications is warranted (such as in creative processes), additional rewards become beneficial because they preserve those divergent judgments rather than averaging them away. The pairwise‑calibration objective provides a direct signal for this decision.
>
> The procedure we describe fits reward functions sequentially. After each new reward is added, we track the pairwise‑calibration loss on a held‑out set and terminate once the validation error no longer decreases meaningfully, so the ensemble grows only until further expansion stops providing benefit. This gives a principled and automated way of choosing the "appropriate" number of reward functions.
>
> > * Computational cost: training and maintaining multiple policies significantly increases resource requirements. Are there strategies to mitigate this overhead?
>
> Indeed, in the current implementation, the compute required for an ensemble of $k$ reward functions scales linearly with $k$, both for training and any downstream fine-tuning or storage of policy variants. Theorem 2 bounds this overhead: the worst‑case number of rewards needed to reach accuracy tolerance $\varepsilon$ grows only linearly in $1/\varepsilon$, so $k$ can remain modest for realistic tolerance levels.
>
> To mitigate this overhead at deployment, we can use approaches like best-of-N sampling or importance sampling that draw candidate completions from a single base model and rescore them with the ensemble. These methods avoid fully fine-tuning multiple model copies while keeping the output policy aligned with the new reward signals.

---

### Comment · Area_Chair_zonr · 2025-08-04

Dear Reviewers,

Please kindly note that the author-reviewer discussion period has started. Please take a look at the rebuttals and acknowledge the receipt of it. Meanwhile, you are encouraged to initiate further discussions with the reviewers.

Best,

Your AC

---

### Decision · Program_Chairs · 2025-09-17

**Decision:**

Accept (poster)

**Comment:**

This work propose a method to align model behavior when human preference can be diverse. Specifically, instead of assuming a ground-truth reward function, the authors allow each annotator to have different reward function, and use a distribution over the reward functions to reflect the disagreement. The authors introduce the concept of epsilon-pairwise calibration that measures the quality of the  distribution. Both theoretical results and empirical evaluations are provided.

The paper is generally well-motivated and the presentation is great. The idea of multiple preference is natural and the results presented in the paper is solid. The reviewers also support the acceptance of the work.